

# Quasiparticles of widely tuneable inertial mass: The dispersion relation of atomic Josephson vortices and related solitary waves

**Sophie S. Shamailov and Joachim Brand***

Dodd-Walls Centre for Photonic and Quantum Technologies, Centre for Theoretical
Chemistry and Physics and New Zealand Institute for Advanced Study, Massey University,
Private Bag 102904 NSMC, Auckland 0745, New Zealand

* J.Brand@Massey.ac.nz

## Abstract

Superconducting Josephson vortices have direct analogues in ultracold-atom physics as solitary-wave excitations of two-component superfluid Bose gases with linear coupling. Here we numerically extend the zero-velocity Josephson vortex solutions of the coupled Gross-Pitaevskii equations to non-zero velocities, thus obtaining the full dispersion relation. The inertial mass of the Josephson vortex obtained from the dispersion relation depends on the strength of linear coupling and has a simple pole divergence at a critical value where it changes sign while assuming large absolute values. Additional low-velocity quasiparticles with negative inertial mass emerge at finite momentum that are reminiscent of a dark soliton in one component with counter-flow in the other. In the limit of small linear coupling we compare the Josephson vortex solutions to sine-Gordon solitons and show that the correspondence between them is asymptotic, but significant differences appear at finite values of the coupling constant. Finally, for unequal and non-zero self- and cross-component nonlinearities, we find a new solitary-wave excitation branch. In its presence, both dark solitons and Josephson vortices are dynamically stable while the new excitations are unstable.



# 1  Introduction

The concept of inertial (or effective) mass [1] is commonly used in condensed matter physics: it captures the response of a quasiparticle in an interacting system to an applied force, encapsulating the emergent Newton's equations of quasiparticle dynamics. Atomic Bose-Einstein condensates (BECs) [2, 3] provide a platform for the emulation of other quantum-many-body systems under highly controllable conditions [4–6]. The possibility of adjusting the inertial mass of localized excitations in BECs by tuning experimental parameters could potentially open the way to interesting applications. Solitary waves with large inertial mass, identified as solitonic vortices [7, 8], have recently been observed in superfluid Fermi gases [9, 10] and BECs [11]. Since the inertial mass of a solitonic vortex depends on the mass density of the superfluid and the length scale of the transverse confinement [10, 12, 13] it is tuneable through the trap geometry within certain limits, but it cannot change sign. In this work we study the properties of solitary waves in a one-dimensional two-component atomic superfluid where an adjustable linear coupling between the two components provides a convenient control parameter that can be used to tune the inverse inertial mass through zero, and thus achieve large positive and negative inertial masses.

Quasi-one-dimensional BECs have been prepared experimentally almost two decades ago [14], and more recently, two coherently-coupled one-dimensional BECs have been demonstrated [15–18]. Dark solitons [19] are localized density depletions with a phase drop across them that propagate at constant speed, preserving their shape. They have been observed in single-component BECs [14, 20–22] and require quasi-one-dimensional confinement to be stable and long-lived [7, 23, 24]. Josephson vortices are known to occur as quantised magnetic flux lines corresponding to a vortex of the superconducting order parameter inside a long Josephson junction between two bulk superconductors [25, 26], and their theoretical descrip-

tion is often reduced to a sine-Gordon equation [27]. Josephson vortices in atomic superfluids were first discussed as domain walls of the relative phase in two-component BECs with a weak coherent coupling between the components [28], and later as vortices that have entered the extended barrier region of a single-component BEC in a double-well geometry [29, 30] (see Fig. 1). While observation of regular vortices is by now common place in BECs [31, 32] and superfluid Fermi gases [33], proposals for the observation of Josephson vortices in BECs were made in [34–36]. Very recently, spontaneously created Josephson vortices were identified through interference patterns in linearly coupled BECs [37].

The model of linearly-coupled two-component BECs describes two different physical realizations: a spinor BEC with two spin components and coherent coupling achieved through radio-frequency or microwave radiation driving a hyperfine transition [28], or a single-component BEC in a double-well potential [6,36,38–40]. Both options are illiustrated in Fig. 1. Theoretical considerations ranged from testing the Kibble-Zurek mechanism in a ring geometry [36], to modeling the decay of an unstable vacuum to a universe with structure [6,41], to metastable domain walls [28,38], to the dynamical response to periodic modulation [39], and to tunneling quenches leading to breather modes forming out of quantum fluctuations [40]. Out of these studies, Refs. [6,28,39,40] reduced the model to the integrable sine-Gordon case, assumed to be applicable in the small tunneling limit, in order to obtain their main results. The experiment [37] has assumed likewise. Testing the validity of this approximation is part of the work presented in the current paper.

After the work presented in this paper was completed we became aware of the related recent work by Qu *et al.* [42], which considers Josephson vortices (called magnetic solitons in [42]) in a regime of weak tunneling and almost equal self- and cross-nonlinearities, where the particle number density is approximately constant.[1] They find analytical and numerical results for solitary wave solutions, their dispersion relations and their dynamics under harmonic trapping. Their findings are consistent with our own results, which cover wider and more general parameter regimes. Furthermore, we have independently obtained similar predictions for the dynamics of the Josephson vortex in a trapped condensate.

In this work we consider several families of solitary nonlinear waves in quasi-one-dimensional linearly-coupled two-component BECs. Son and Stephanov [28] discussed the existence of a domain-wall of the relative phase in this system, which was later called a Bose Josephson vortex by Kaurov and Kuklov [29, 30]. The latter authors found exact stationary solutions to the coupled Gross-Pitaevskii equations and discussed the bifurcation of the dark soliton solution of this model, which is stable above a critical strength of the linear coupling, into stable Josephson vortices which coexist at smaller coupling strength with the now unstable dark soliton. Qadir *et al*. [44] added a detailed stability analysis and approximated the properties of moving Josephson vortices for small velocities. Exact solutions and the full dispersion relation for moving Josephson vortices have so far not been available and approximate dispersion relations have only been found for the regime of weak coherent coupling and nearly-equal nonlinear interactions [42]. Here we present numerical results for the complete dispersion relation of solitary wave solutions including moving Josephson vortices (already used in Ref. [43] to simulate collisions of Josephson vortices). In addition to the stable Josephson-vortex solutions, a new unstable branch of solitary waves arises at finite cross-component nonlinear interactions where both dark solitons and Josephson vortices are stable.

We show that there is a critical value of the linear coupling at which the Josephson vortex dispersion relation changes from having a single maximum to having three (local) extrema: a maximum, minimum and another maximum. At this point the inertial mass of the Josephson vortex at the center of the dispersion relation changes sign and diverges to $\pm\infty$ on either

---

[1]The main results of our work were first presented at the Australasian Workshop on Emergent Quantum Matter 2014, and used as input for Ref. [43].

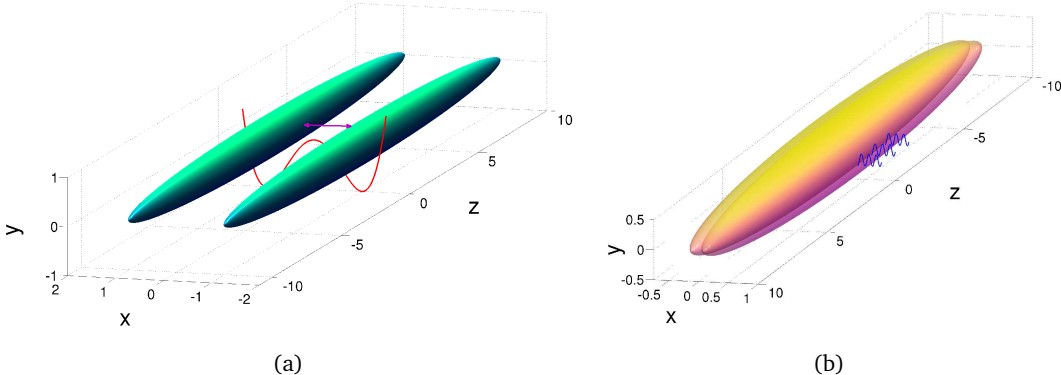

Figure 1: Possible realisations of a linearly-coupled two-component Bose gas as described by Eqs. (1) and (6). (a) A scalar Bose gas confined in an elongated double-well potential. A tight binding approximation leads to Eq. (1), where $J$ describes tunneling through the potential barrier and the cross-interaction vanishes ($g_c = 0$, $\gamma = 1$). (b) An atomic Bose gas with two accessible (hyperfine) spin components in a cigar-shaped quasi-one-dimensional trap. The two components are slightly off-shifted for clarity. Linear coupling of the spin-components with constant $J$ is achieved by coherent driving with a radio-frequency field shown in blue. In this case, the cross-component interaction $g_c$ is typically of similar magnitude to $g$ ($g_c \approx g$, $|\gamma| \ll 1$) but can be tuned to different values by means of a Feshbach resonance for certain atomic species [41,45].

side of the bifurcation. The two maxima appearing at smaller coupling strength correspond to small-velocity solitary waves with negative inertial mass that resemble a dark soliton in one component with a back-flow current in the other.

In the small coupling limit, we test whether the central part of the Gross-Pitaevskii Josephson vortex dispersion relation approaches the sine-Gordon dispersion relation. The latter can be described by only two parameters – the "mass" and the "speed of light". The inertial mass of the Josephson vortex approaches the sine-Gordon "mass" parameter quite rapidly as the tunneling is decreased, but the "speed of light" does not – the approach to the common value at zero tunneling occurs with different slopes.

The paper is structured as follows. Section 2 introduces the model equations, 3 defines useful observables for characterizing the solutions, and section 4 summarizes known analytical solutions and their properties. Next, section 5 explains how we numerically obtain solutions, which are visualized in section 6. Section 7 discusses the dispersion relations, while 8 discusses parameter regimes and a stability analysis of the solutions. Section 9 addresses the inertial mass and missing particle number of Josephson vortices. Section 10 presents a variational approximation to slow-moving Josephson vortices, yielding analytical expressions for the key numerical results. Section 11 summarizes some central results regarding the sine-Gordon equation, and 12 examines the validity of approximating the Gross-Pitaevskii equations with the sine-Gordon model. Discussion and conclusions are given in 13. Appendix A presents details of how we perform the stability calculation and appendix B derives the sine-Gordon equation from the Gross-Pitaevskii model, thus enabling a direct comparison of the two.

## 2  The model

We are considering a quasi-one-dimensional two-component Bose gas with identical boson mass $m$ for the two components and linear coupling $J$ in the Gross-Pitaevskii approximation

with the energy functional

$$W = \int \left\{ \sum_{j=1,2} \left[ \frac{\hbar^2}{2m} \left| \partial_x \Psi_j \right|^2 + \frac{g}{2} |\Psi_j|^4 - \mu |\Psi_j|^2 \right] + g_c |\Psi_1|^2 |\Psi_2|^2 - J \left( \Psi_1^* \Psi_2 + \Psi_2^* \Psi_1 \right) \right\} dx, \quad (1)$$

where $\Psi_j(x, t)$ is the order parameter of component $j \in \{1, 2\}$, $\mu$ is the common chemical potential and for simplicity, the intra-component interaction constant $g$ [46] is also taken as common for the two components, but the cross-component interaction constant $g_c$ may be different. We assume that $J > 0$, which is the physical case when the coupling is achieved by tunnelling through a potential barrier [47]. In the general case, the phase of $J$ can be absorbed into the definition of $\Psi_2$. Two possible realisations of this model are illustrated in Fig. 1. Even though the homogeneous one-dimensional Bose gas never completely fully condenses, the Gross-Pitaevskii (mean-field) approximation is justified when $mg/\hbar^2 \tilde{n}_0 \ll 1$, i.e. the Lieb-Linger parameter is small [48], where $\tilde{n}_0 = (\mu + J)/(g + g_c)$ is the background particle density in each component.

The time evolution of the order parameter is described by the Gross-Pitaevskii equation, which can be formally obtained from $i\hbar \partial_t \Psi_j = \delta W[\Psi_j^*, \Psi_j]/\delta \Psi_j^*$:

$$i\hbar \partial_t \Psi_1 = -\frac{\hbar^2}{2m} \partial_{xx} \Psi_1 - \mu \Psi_1 + g |\Psi_1|^2 \Psi_1 + g_c |\Psi_2|^2 \Psi_1 - J \Psi_2,$$
$$i\hbar \partial_t \Psi_2 = -\frac{\hbar^2}{2m} \partial_{xx} \Psi_2 - \mu \Psi_2 + g |\Psi_2|^2 \Psi_2 + g_c |\Psi_1|^2 \Psi_2 - J \Psi_1. \quad (2)$$

We are interested in solitary wave solutions that translate with a constant velocity $V_s$. With the aim of adimensionalising the equations of motion we make the ansatz

$$\Psi_j(x, t) = \sqrt{\frac{\mu}{g + g_c}} \psi(z), \quad (3)$$

where

$$z = \xi - v_s \tau,$$
$$\xi = \frac{\sqrt{m\mu}}{\hbar} x,$$
$$\tau = \frac{\mu}{\hbar} t,$$
$$v_s = \sqrt{\frac{m}{\mu}} V_s, \quad (4)$$

are dimensionless variables. Further introducing the dimensionless linear and nonlinear coupling constants

$$\nu = \frac{J}{\mu},$$
$$\gamma = \frac{g - g_c}{g + g_c}, \quad (5)$$

we obtain the dimensionless Gross-Pitaevskii equations for uniformly translating solutions

$$-iv_s \partial_z \psi_1 = -\frac{1}{2} \partial_{zz} \psi_1 - \psi_1 + \frac{1}{2}(1 + \gamma) |\psi_1|^2 \psi_1 + \frac{1}{2}(1 - \gamma) |\psi_2|^2 \psi_1 - \nu \psi_2,$$
$$-iv_s \partial_z \psi_2 = -\frac{1}{2} \partial_{zz} \psi_2 - \psi_2 + \frac{1}{2}(1 + \gamma) |\psi_2|^2 \psi_2 + \frac{1}{2}(1 - \gamma) |\psi_1|^2 \psi_2 - \nu \psi_1. \quad (6)$$

We are looking for solutions to these equations that asymptotically tend to the stable constant background solution for $z \to \pm\infty$ [47]. The boundary conditions can thus be written as

$$
\lim_{z\to\pm\infty} \psi_1 = \lim_{z\to\pm\infty} \psi_2,
$$
$$
\lim_{z\to\pm\infty} |\psi_1|^2 = \lim_{z\to\pm\infty} |\psi_1|^2 = 1 + \nu. \tag{7}
$$

Note that this leaves a complex phase undetermined at each end, and while Eqs. (6) are invariant under the change of an overall phase, the phase difference

$$
\Delta\phi = \arg \frac{\psi_j(z \to -\infty)}{\psi_j(z \to \infty)}, \tag{8}
$$

bears physical significance.

## 3  Physical observables

Let us now define several useful quantities that shall be evaluated later on for the numerical solutions. The energy functional in dimensionless units is evaluated as

$$
E = \frac{\sqrt{\mu m}(g + g_c)}{\hbar \mu^2} W =
$$
$$
= \int_{-L}^{L} dz \sum_{k=1,2} \left\{ \frac{1}{2} |\partial_z \psi_k|^2 - |\psi_k|^2 - \nu \psi_k^* \psi_{3-k} + \frac{1}{4}(1+\gamma)|\psi_k|^4 \right\} + \frac{1}{2}(1-\gamma)|\psi_1|^2|\psi_2|^2. \tag{9}
$$

A key quantity is the excitation energy associated with the solitary wave

$$
E_s = E_{\text{sol}} - E_0, \tag{10}
$$

where $E_0 = -2L(1 + \nu)^2$ is the (dimensionless) energy of the constant background, and $E_{\text{sol}}$ is the energy of the solitary wave solution.

For a localised solitary wave solution that heals to the constant background, $E_s$ is independent of the box size $2L$ for sufficiently large $L$. It will depend on the soliton velocity $v_s$ but there may be multiple solutions for each $v_s$.

Another useful observable is the momentum, which is scaled by $\hbar \mu/(g + g_c)$. The background solution has zero momentum and we introduce the following dimensionless observables:

$$
\begin{aligned}
P_s &= \int_{-L}^{L} (p_1 + p_2)\, dz, \\
\Delta P &= \int_{-L}^{L} (p_1 - p_2)\, dz, \\
p_k &= -\frac{i}{2}\left[ \psi_k^* \frac{d\psi_k}{dz} - \psi_k \frac{d\psi_k^*}{dz} \right], \\
P_{cf} &= 2(1 + \nu)\Delta\phi, \\
P_c &= P_s + P_{cf},
\end{aligned} \tag{11}
$$

where $P_s$ is the physical momentum of the solitary wave with boundary conditions (7). The momentum difference $\Delta P$ between the two components indicates a degree of symmetry breaking. In the scenario where the two components are spatially separated it has the significance of an orbital angular momentum (see Fig. 1). The quantity $P_{cf}$ is the momentum of the counter flow that has to be added in periodic boundary conditions (ring geometry) in order to compensate for the phase step $\Delta\phi$. The canonical momentum $P_c$ is the momentum that the solitary wave excitation has with periodic boundary conditions but it is also significant for the open boundary conditions (7) due to the relation

$$\frac{dE_s}{dP_c} = v_s. \tag{12}$$

The canonical momentum provides a convenient way of parameterising the solitary-wave solutions and the relation of $E_s$ vs. $P_c$ is known as the dispersion relation. Examples of dispersion relations that summarise the results of this work are presented in section 7, Fig. 5, panels (a), (c), (e). In the framework of Landau's quasiparticle picture, the dispersion relation determines the dynamics of the solitary waves in a slowly-changing environment as long as the nature of the solitary wave changes adiabatically such that the solitary wave solutions are well approximated by the stationary solutions of Eqs. (6) at any one time and the energy stored in the solitary wave is conserved [12,49].

Of particular interest is the inertial mass of the quasiparticle, which is given by

$$m_I = \frac{dP_c}{dv_s} = 2\frac{dE_s}{d(v_s^2)} = \left(\frac{d^2 E_s}{dP_c^2}\right)^{-1}. \tag{13}$$

It is directly related to the measurable oscillation frequency $\Omega$ of the solitary wave under the influence of a weak harmonic trap in the longitudinal direction with frequency $\omega_z$ by

$$\frac{\omega_z^2}{\Omega^2} = \frac{m_I}{m_P}, \tag{14}$$

where $m_P$ is the physical mass [12,50–52]. At zero velocity, the physical mass is proportional to the particle number depletion of the solitary wave by $m_P = mN_d|_{v_s=0}$.[2] The (missing) particle number of the solitary wave $N_d$ is obtained by integrating the density and subtracting the background

$$N_d = \int_{-L}^{L} [n_1(z) + n_2(z) - 2n_0]\, dz, \tag{15}$$

where $n_k(z) = |\psi_k(z)|^2$ are the dimensionless particle densities in the two components and $n_0 = 1 + \nu = \tilde{n}_0(g + g_c)/\mu$ is the dimensionless background density. Note that Eq. (15) yields the particle number in terms of the reduced dimensionless quantities and is scaled by a factor $\hbar\mu/\sqrt{\mu m}(g + g_c)$.

## 4 Analytically-known solitary-wave solutions

Several exact solutions of (6) are known. The lowest-energy constant solution is $\psi_1 = \psi_2 = \sqrt{1 + \nu}$, which we shall refer to as the background. We remark that $\gamma = 0$ separates the miscible ($\gamma > 0$) and immiscible ($\gamma < 0$) phases of the system. In the miscible regime, $\nu = 0$

---

[2]Equation (14) can be derived from defining $m_P = -m\,dE_s/d\mu$ [51,52]. The general relation between $m_P$ and $N_d$ will be discussed elsewhere [53].

corresponds to a degenerate mean-field ground state with undefined spin polarisation, with any $v \neq 0$ leading to a background solution polarised along the $x$-direction. For this work we consider the miscible regime with $\gamma \geq 0$ and a polarised ground state along the $+x$-direction obtained with $v > 0$, while analogous results hold for $v < 0$, where polarisation along the $-x$-direction is obtained.

## 4.1 Dark solitons

The coupled BECs system supports dark soliton solutions which satisfy $\psi = \psi_1 = \psi_2$ and are given by [1]

$$\psi = \sqrt{1 + v - v_s^2} \tanh\left[\sqrt{1 + v - v_s^2}z\right] + iv_s. \tag{16}$$

This corresponds to identical dark soliton solutions in each component with $-v_B \leq v_s \leq v_B$. The maximal velocity at which a dark soliton can travel is the Bogoliubov speed of sound of the system, $v_B = \sqrt{1 + v}$. Note that the dark soliton solutions are independent of the cross-interaction parameter $\gamma$. The zero-velocity case is visualised in Fig. 2 (a) & (b).

The soliton's properties can be calculated by direct integration from the analytical solution and are well known [1]. In our dimensionless units they explicitly depend on the coupling parameter $v$. The excitation energy

$$E_s = \frac{8}{3}\left(1 + v - v_s^2\right)^{3/2}, \tag{17}$$

takes a maximum value at zero velocity and vanishes at $v_s = \pm v_B$. The phase difference

$$\Delta\phi = \pi - 2\tan^{-1}\left[\frac{v_s}{\sqrt{1 + v - v_s^2}}\right] \tag{18}$$

is $\pi$ for stationary solitons and reaches the extremal values 0 and $2\pi$ at the limiting velocities $\pm v_B$. The missing particle number of the dark soliton evaluates to

$$N_d = -4\sqrt{1 + v - v_s^2}, \tag{19}$$

and the momentum difference vanishes ($\Delta P = 0$). The velocity dependence of the dark soliton's properties is shown in section 7, Figs. 6–8 as green lines.

The canonical momentum of the dark soliton is

$$P_c = 2\pi(1 + v) - 4v_s\sqrt{1 + v - v_s^2} - 4(1 + v)\tan^{-1}\left(\frac{v_s}{\sqrt{1 + v - v_s^2}}\right), \tag{20}$$

varying in the interval $P_c \in [0, 4\pi(1 + v)] = [0, 2\pi n_0]$. The inertial mass evaluates to $m_I = -8\sqrt{1 + v}$.

## 4.2 Stationary Josephson vortex

The stationary Josephson vortex is found as a complex solution of Eq. (6), which breaks the symmetry between the two components [28, 29]. It is given by $\psi_1 = \psi_2^* = \psi$, with

$$\psi = \sqrt{1 + v}\tanh\left(2\sqrt{v}z\right) + i\sqrt{1 - 3v}\,\operatorname{sech}\left(2\sqrt{v}z\right), \tag{21}$$

and only exists for $0 < v < \frac{1}{3}$. The parameter value $v = \frac{1}{3}$ marks a bifurcation point where the Josephson vortex solution becomes identical to the dark soliton solution (16). For $v < \frac{1}{3}$ two degenerate solutions are obtained from $\psi$ and $\psi^*$, which can be interpreted as vortices of opposite circulation. The vortex nature is most clearly seen in the double-well potential

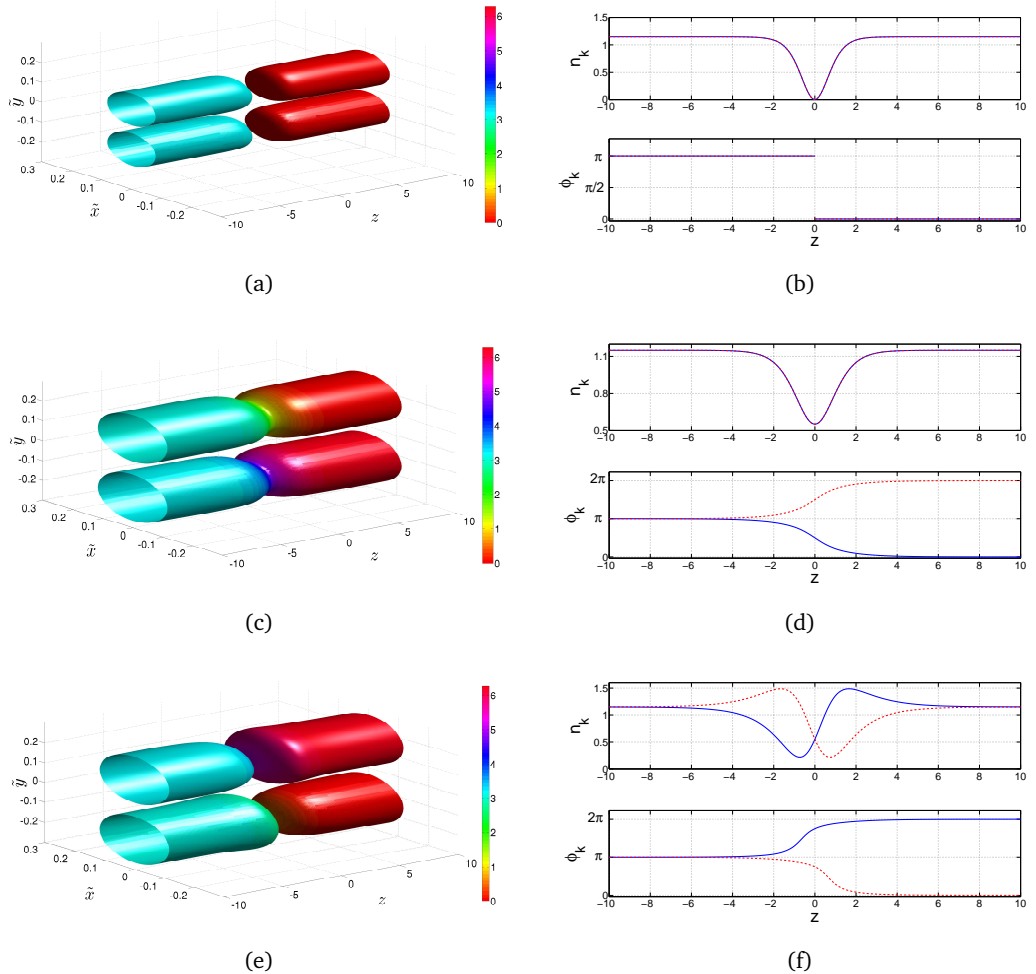

Figure 2: Stationary solitary-wave solutions ($v_s = 0$) in a coupled two-component BEC: (a) & (b) Dark soliton, (c) & (d) Josephson vortex, and (e) & (f) Manakov soliton at $\theta = -\pi/4$. Left column: The density $n_k(z) = |\psi_k(z)|^2$ is encoded in the width of the tube and the phase $\arg[\psi_k(z)]$ in the colour. The upper tube corresponds to component 1 and the lower one to component 2. Specifically we define $N_k(z, \tilde{x}, \tilde{y}) = n_k(z) \exp[-\frac{(\tilde{y} \pm 0.15)^2 + \tilde{x}^2}{0.1^2}]$ and plot isosurfaces at the value $N_k = 0.3$. Right column: density (top panels) and phase (bottom panels) profiles are shown directly as a blue dashed line (component 1) and a red dashed line (component 2). Other parameters are $\gamma = 1, v = 0.15, P_c = 2\pi(1 + v)$ for all plots.

scenario of Fig. 1 (a), where a phase singularity sits in the middle of the double-well barrier at $z = 0$. Indeed, tracing the phase along the $+z$ direction in $\psi_1$ and along $-z$ in $\psi_2$ amounts to a total phase change of $2\pi$ if the origin $z = 0$ is included; note that both components have equal phase far away from the Josphson vortex. This can be seen following the phase profiles shown in Fig. 2 (c) & (d).

The energy and momentum difference for the Josephson vortex at $v_s = 0$ are

$$E_s = \frac{8}{3}\sqrt{v}(3 - v),$$
$$\Delta P = \mp 2\pi\sqrt{1 + v}\sqrt{1 - 3v}, \tag{22}$$

where the $-(+)$ stands for the Josephson vortex $\psi$ (anti-vortex $\psi^*$).

## 4.3  Manakov solitons

When the cross- and intra-component nonlinearities are equally strong (*i.e.* $\gamma = 0$), the coupled equations (6) can be mapped on to the integrable vector non-linear Schrödinger equation known as the Manakov system [54, 55]. In this limit, a whole family of solutions can be found analytically [56]. Defining $\chi_{1,2} = \frac{1}{\sqrt{2}}(\psi_2 \pm \psi_1)$, we re-write equations (6) for the new variables:

$$-i\nu_s \partial_z \chi_k = -\frac{1}{2}\partial_{zz}\chi_k - (1 \pm \nu)\chi_k + \frac{1}{2}\left(|\chi_1|^2 + |\chi_2|^2\right)\chi_k, \tag{23}$$

where the two different signs in front of $\nu$ are to be taken with the two different indices, $k = 1, 2$. A trial solution of the form [56]

$$\begin{aligned}
\chi_1 &= \alpha i + \beta \tanh(\eta z), \\
\chi_2 &= \delta \operatorname{sech}(\eta z)e^{i\varepsilon z}
\end{aligned} \tag{24}$$

is found to satisfy (23) if the parameters are given by

$$\begin{aligned}
\alpha &= \sqrt{\frac{1+\nu}{2\nu}}\nu_s, \\
\beta &= \sqrt{\frac{(4\nu - \nu_s^2)(1+\nu)}{2\nu}}, \\
\eta &= \sqrt{4\nu - \nu_s^2}, \\
\delta &= \sqrt{\frac{(4\nu - \nu_s^2)(1-3\nu)}{2\nu}}, \\
\varepsilon &= \nu_s.
\end{aligned} \tag{25}$$

In fact, $\chi_2$ may be multiplied by an arbitrary phase factor, $e^{i\theta}$, and the resulting solution still satisfies the differential equations. Transforming back to the $\psi$-fields gives

$$\psi_{1,2} = \frac{1}{\sqrt{2}}\left[\alpha i + \beta \tanh(\eta z) \pm e^{i\theta}\delta \operatorname{sech}(\eta z)e^{i\varepsilon z}\right]. \tag{26}$$

Notice that for the parameters in (25) to be real (and the solution to be non-trivial) we need $\nu < 1/3$ and $\nu_s^2 < 4\nu$. The phase angle $\theta$ remains a free parameter, indicating the large degeneracy of these solutions. For $\nu_s = 0$ and $\theta = \mp\pi/2$ the solution (26) reduces to the stationary Josephson vortex (anti-vortex). We will refer to the family of solutions (26) as the Manakov solutions, even though the presence of the linear coupling $\nu$ provides a point of difference to the solutions of the original Manakov system.

As for the dark soliton, it is possible to calculate all the quantities of interest for the Manakov solutions at $\gamma = 0$ analytically: the excitation energy, angular momentum, phase difference, canonical momentum, and missing particle number are

$$\begin{aligned}
E_s &= 4\sqrt{4\nu - \nu_s^2}\left[\frac{2}{3}(4\nu - \nu_s^2) - (3\nu - 1)\right], \\
\Delta P &= 2\pi\sqrt{1+\nu}\sqrt{1-3\nu}\,\operatorname{sech}\left(\frac{\pi\nu_s}{2\sqrt{4\nu - \nu_s^2}}\right)\sin\theta, \\
\Delta\phi &= \pi - 2\tan^{-1}\left[\frac{\nu_s}{\sqrt{4\nu - \nu_s^2}}\right], \\
P_c &= 2\pi(1+\nu) - 4\left\{\nu_s\sqrt{4\nu - \nu_s^2} + (1+\nu)\tan^{-1}\left[\frac{\nu_s}{\sqrt{4\nu - \nu_s^2}}\right]\right\}, \\
N_d &= -4\sqrt{4\nu - \nu_s^2}.
\end{aligned} \tag{27}$$

The inertial mass evaluates to $m_I = -2\frac{5\nu+1}{\sqrt{\nu}}$ and is independent of velocity. Note that the limits of $P_c$ are the same as for dark solitons. The Manakov solitons are illustrated in Fig. 2 (e) & (f).

# 5 Numerical methods

In order to extend the analytical solutions into unknown parameter regimes we numerically solve the boundary value problem with open boundary conditions and $z \in [-L, L]$ as described in section 2.[3] As boundary conditions, we require zero first derivatives at $\pm L$ for both fields and choose $L$ large enough for the solutions to settle in to the constant background.

After a solution is obtained, we check that the densities $n_k$ at $\pm L$ are within 0.01 of the background density, and that the phases of the two fields at $\pm L$ are within 0.01 of each other ($\phi_k(\pm L) = \phi_{3-k}(\pm L)$). If either condition is not fulfilled, $L$ is increased and the solver is called again.

For the numerical procedure, an appropriate guess for the wave-function has to be provided. The initial guess is obtained from one of the analytically-known solutions and is then followed in one of the parameters. All parts of the dispersion relation could be conveniently accessed by changing either of the controlling parameters $v_s$, $\nu$, $\gamma$ in small steps.

We found that out of the entire $\theta$-spectrum of analytic Manakov solutions at $\gamma = 0$, only the $\theta = 0, \pm\pi$ and $\theta = \pm\pi/2$ solutions extend to positive, finite $\gamma$. When $\gamma = 0$, the stationary Manakov solution is identical to the zero-velocity Josephson vortex solution if $\theta = -\pi/2$. Indeed, following the $\theta = -\pi/2$ solution from $\gamma = 0$ to $\gamma > 0$ yields the Josephson vortex branch obtained by following Josephson vortices from $\gamma = 1$ to $\gamma < 1$. On the other hand, following the $\theta = 0$ solution from $\gamma = 0$ to $\gamma > 0$ gives an entirely new branch, which we shall refer to as *staggered solitons*, due to the fact that the centers of the density dips are shifted with respect to each other (see Fig. 4 (c)-(f)).

# 6 Visualizing the solutions

In order to visualize the solutions, we show surface plots where the width of the two cylinders is related to the density of the two fields and the phase is encoded as a colour map. In addition, we provide one-dimensional plots of the density and phase profiles to better resolve the finer details. We choose representative examples that illustrate the different solutions in all distinct regions of parameter space.

Figure 3 (a) & (b) show a moving Josephson vortex for $\gamma = 1, \nu = 0.15$. In all cases for $\gamma = 1, \nu \geq 0.15$ the solutions were obtained by starting from the known zero-velocity Josephson vortices (21) and increasing velocity at a fixed $\nu$. Note that physically, at $P_c = 2\pi(1 + \nu), v_s = 0$, the Josephson vortex is centered exactly half way between the two parallel BEC lines. Its distinctive features are an equal dip in the density and an equal-but-opposite phase step in each condensate.

Figure 3 (c) & (d) show a stationary Josephson vortex at the maximum of the dispersion relation for $\gamma = 1, \nu = 0.005$ and panels (e) & (f) show a moving Josephson vortex for the same $\gamma$ and $\nu$. The solutions for $\gamma = 1, \nu < 0.15$ were obtained by starting from the previously-calculated wavefunctions at $\nu = 0.15$, and at each velocity gradually decreasing $\nu$.

Note that, as shown in section 7, Fig. 5 (a), as $\nu$ goes to zero, the Josephson vortex dispersion relation is "split in half" as $E_s(P_c = 2\pi(1 + \nu))$ drops to zero. At $\nu = 0$ each "wing" of the dispersion relation corresponds to a dark soliton in one of the two BEC lines. This can be seen

---

[3]We use the boundary value problem solver `bvp5c.m` from the MATLAB environment, with the absolute and relative tolerances set to $10^{-8}$.

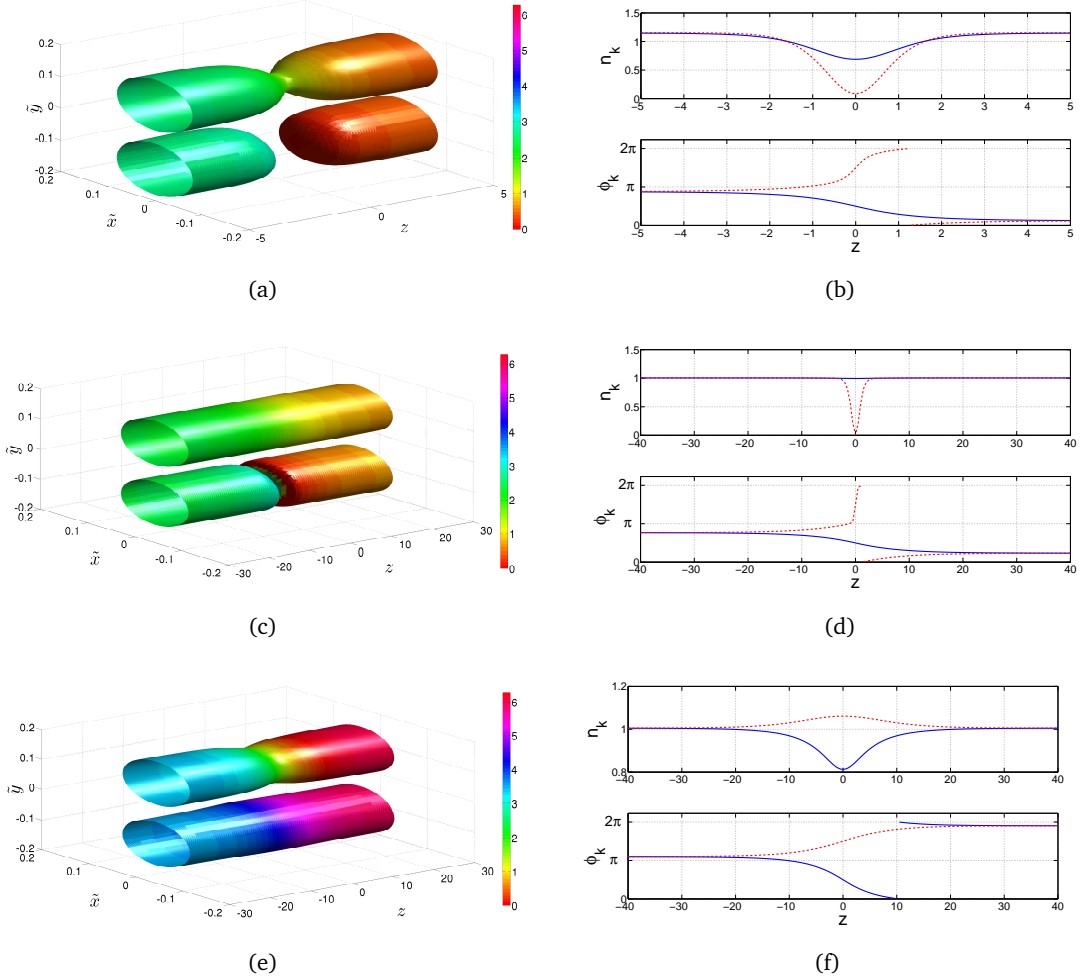

Figure 3: Numerical solutions from the Josephson vortex family corresponding to various points of interest on the dispersion relations of Fig. 5. Left column: The density $n_k(z) = |\psi_k(z)|^2$ is encoded in the width of the tube and the phase $\arg[\psi_k(z)]$ in the colour. The upper tube corresponds to component 1 and the lower one to component 2. Specifically we define $N_k(z, \tilde{x}, \tilde{y}) = n_k(z) \exp[-\frac{(\tilde{y}\pm 0.1)^2 + \tilde{x}^2}{0.1^2}]$ and plot isosurfaces at the value $N_k = 0.6$. Right column: density (top panels) and phase (bottom panels) profiles are shown directly as a blue dashed line (component 1) and a red dashed line (component 2). (a) & (b) Moving Josephson vortex: $\gamma = 1, v = 0.15, P_c = 1.34\pi$, (c) & (d) stationary Josephson vortex maximum: $\gamma = 1, v = 0.005, P_c = 1.05\pi$, (e) & (f) moving Josephson vortex: $\gamma = 1, v = 0.005, P_c = 2.17\pi$.

clearly in Fig. 3 (c) & (d) where the density of one condensate is practically flat and the other has a strong dip. We therefore refer to the quasi-particles around the maxima of the Josephson vortex dispersion relation as "Josephson vortex maxima", and interpret them as single-strand dark solitons. At the maxima of the dispersion relation, the vortex is exactly crossing one of the BEC strands as it moves out (perpendicularly to the BECs) from in between the two strands. Conversely, the dark soliton dispersion relation consists of a dark soliton in each of the BEC strands and the Josephson vortex dispersion relation merges with it as $v \to 1/3$.

Figure 4 (a) & (b) show an example of a moving Josephson vortex for $\gamma = 0.5, v = 0.005$. The solutions for $v = 0.005, \gamma < 1$ were obtained by starting from the previously-calculated wavefunctions at $v = 0.005, \gamma = 1$, and at each velocity gradually decreasing $\gamma$. Once part

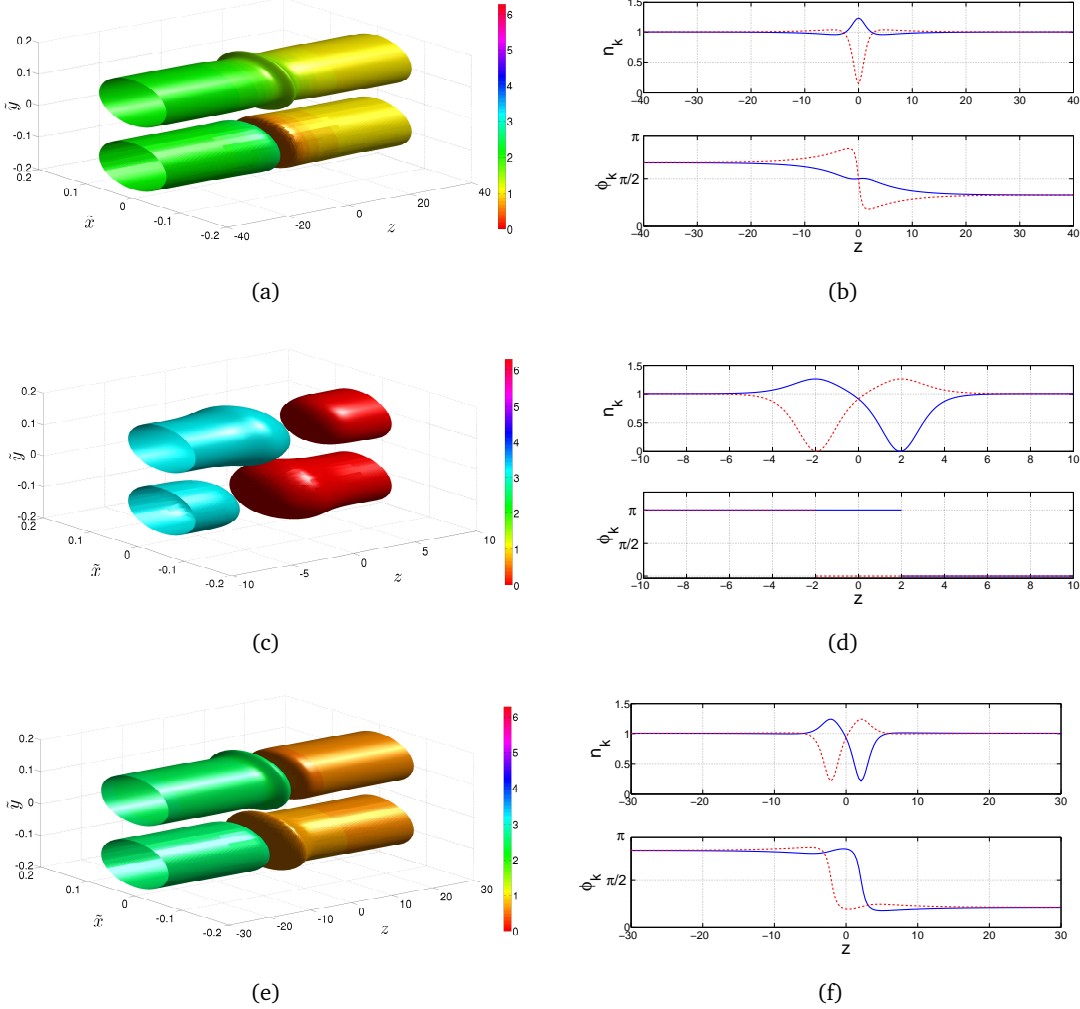

Figure 4: Numerical solutions from the Josephson vortex family corresponding to various points of interest on the dispersion relations of Fig. 5. Left column: The density $n_k(z) = |\psi_k(z)|^2$ is encoded in the width of the tube and the phase $\arg[\psi_k(z)]$ in the colour. The upper tube corresponds to component 1 and the lower one to component 2. Specifically we define $N_k(z, \tilde{x}, \tilde{y}) = n_k(z) \exp[-\frac{(\tilde{y} \pm 0.1)^2 + \tilde{x}^2}{0.1^2}]$ and plot isosurfaces at the value $N_k = 0.6$. Right column: density (top panels) and phase (bottom panels) profiles are shown directly as a blue dashed line (component 1) and a red dashed line (component 2). (a) & (b) moving Josephson vortex: $\gamma = 0.5, v = 0.005, P_c = 0.49\pi$, (c) & (d) stationary staggered soliton: $\gamma = 0.5, v = 0.005, P_c = 2.01\pi$, (e) & (f) moving staggered soliton: $\gamma = 0.5, v = 0.005, P_c = 0.86\pi$.

of the dispersion relation was available at each $\gamma$ value, if necessary, we could complete it by following in $v_s$.

Figure 4 (c) & (d) show a stationary staggered soliton for $\gamma = 0.5$, $v = 0.005$ and panels (e) & (f) show a moving staggered soliton for the same $\gamma$ and $v$. These solutions were obtained by starting from the analytical Manakov wavefunctions at $v = 0.005, \gamma = 0$, and at each velocity gradually increasing $\gamma$. This gave us the central part of the dispersion relation at all $\gamma$ values, which we then extended in $v_s$ at each constant $\gamma$.

## 7  Dispersion relation and other observables

Figure 5 panels (a), (c), (e) show the dispersion relations of dark solitons, Josephson vortices and of the staggered solitons, the latter only for $\gamma < 1$. From panel (a) it is clear that for $\gamma = 1$, the Josephson vortex dispersion relation changes concavity at $P_c = 2\pi(1 + v)$ at around $v \approx 0.14 - 0.15$. The same process is observed in reverse as $\gamma \to 0^+$ with $v \leq 0.14$, as we move from panel (c) to (e). At $\gamma = 0$, the equations reduce to the Manakov case, which is solved analytically in section 4.3 and indeed the Manakov solitons have a dispersion relation with a single central maximum.

As is clear from Fig. 5 (a), when $\gamma = 1$ the Josephson vortex dispersion relation bifurcates from the dark soliton one at some critical velocity that depends on $v$, examined later in Fig. 9 (a). In panel (b) for $0 < \gamma < 1$ the staggered soliton branch is unstable and lies between the stable dark soliton & Josephson vortex branches.

In Fig. 5 panels (b), (d), (f) we compare the energy of dark solitons, Josephson vortices, Josephson vortex maxima and staggered solitons at the extrema of the dispersion relations (which necessarily implies zero velocity) as a function of $v$. In (b), for $\gamma = 1$, the Josephson vortex and Josephson vortex maximum lines merge at around $v \approx 0.14 - 0.15$. It may be expected that this bifurcation point depends on $\gamma$, and this is indeed found to be the case. Panel (d) shows that at $\gamma = 0.5$, the bifurcation point has now moved from $v = 0.1413$ to around $v = 0.1$. Notice that the staggered solitons bifurcate from the dark solitons, which accounts for their similar properties. Finally, at $\gamma = 0$ in panel (f), only the dark soliton-Josephson vortex bifurcation remains: Josephson vortices join the dark soliton line at $v = 1/3$, which is independent of $\gamma$.

Note that in (d), both the Josephson vortex and the Josephson vortex maximum solutions are stable, but the Josephson vortex maxima have $P_c \neq 2\pi(1 + v)$, unlike all other solutions shown. The staggered soliton solutions only exist below about $v = 0.125$ where they are unstable, while the higher energy dark solitons are stable. For $v > 0.125$, staggered solitons disappear and dark solitons become unstable.

The energy, missing particle number, angular momentum and phase difference are plotted as a function of velocity in Figs. 6-8 for the three parameter sets that were used in Figs. 3 & 4. The colour code is identical to that used in Fig. 5. The change in concavity of the Josephson vortex branch as $v$ goes down through $v \approx 0.14 - 0.15$ in Fig. 5 (a) is seen as the development of a loop in velocity-energy plots (compare panels (a) of Figs. 6 and 7). In Fig. 8, we see that staggered solitons and Josephson vortices merge and terminate at common end points. Only Josephson vortices have non-zero $\Delta P$ (and are therefore identified as vortices), while dark and staggered solitons have $\Delta P = 0$, and are hence classified as solitons.

## 8  Parameter regimes, types of excitations and their stability

Dark soliton solutions are analytically known for all parameter values. We have numerically obtained all translating Josephson vortex solutions in two parameter regimes: $\gamma = 1$, $0.005 \leq v \leq 0.33$ and $v = 0.005$, $0 \leq \gamma \leq 1$. Staggered solitons were found in the second regime; this branch always has zero angular momentum and energy higher than Josephson vortices but lower than dark solitons. It is understood to be a transitory state through which Josephson vortices are able to reverse their circulation. Wherever the staggered soliton branch does not exist, dark solitons perform the role of the transitory state.

When $\gamma = 1$, Kaurov and Kuklov [29] found that zero-velocity Josephson vortex solutions only exist for $v < 1/3$, at which point Josephson vortices merge into dark solitons. In fact, a bifurcation exists for all velocities, but the critical value of $v$ depends on velocity. A more

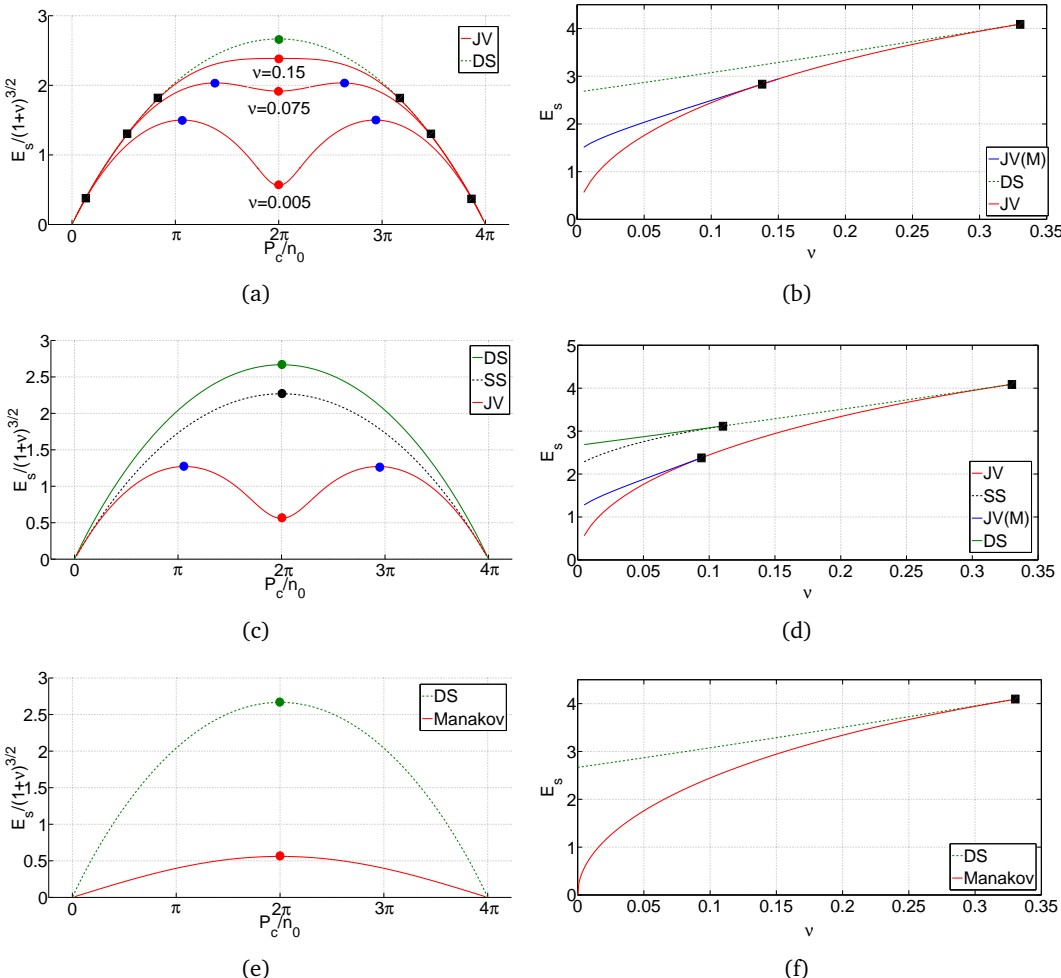

Figure 5: Families of solitary-wave solutions. Left column: dispersion relations with excitation energy $E_s$ vs. canonical momentum $P_c$ for $v = 0.005$ [or as indicated in panel (a)]. Right column: $E_s$ vs. the coupling parameter $v$ for stationary solitary waves ($v_s \equiv dE_s/dP_c = 0$, marked by dots in the left column). (a), (b): $\gamma = 1$ (no cross-interaction, $g_c = 0$). The highest energy branch corresponds to dark soliton solutions (green, labeled 'DS'); the Josephson vortex (red, labeled 'JV') corresponds to the lowest available energy state for given $P_c$. Different values of $v$ are shown in panel (a). For $v < 0.1413$ three local extrema coexist, each corresponding to a stationary solitary wave solution. There are two degenerate maxima (blue, labeled 'JV(M)'), and one minimum corresponding to the stationary Josephson vortex solution. (c), (d): $\gamma = 0.5$. Staggered soliton solutions (unstable) appear at intermediate energies for $0 < \gamma < 1$ (black, labeled 'SS'). (e), (f): $\gamma = 0$. Josephson vortices and staggered solitons merge into the highly degenerate branch of "Manakov" solutions (stable). Throughout: full (dashed) lines indicate stable (dynamically unstable) solutions; square markers indicate bifurcation points.

natural point of view for us will be to say that for any given value of the tunnelling strength $v$, the Josephson vortex and dark soliton dispersion relations merge smoothly at some critical momentum (associated with some critical velocity), and for larger momenta, the Josephson vortex branch does not exist. This is illustrated in Fig. 9 (a) where we plot the maximal velocity reached by the Josephson vortex branch (the critical velocity) as a function of $v$. We found that, with $\gamma = 1$, whenever Josephson vortices and dark solitons coexist, Josephson

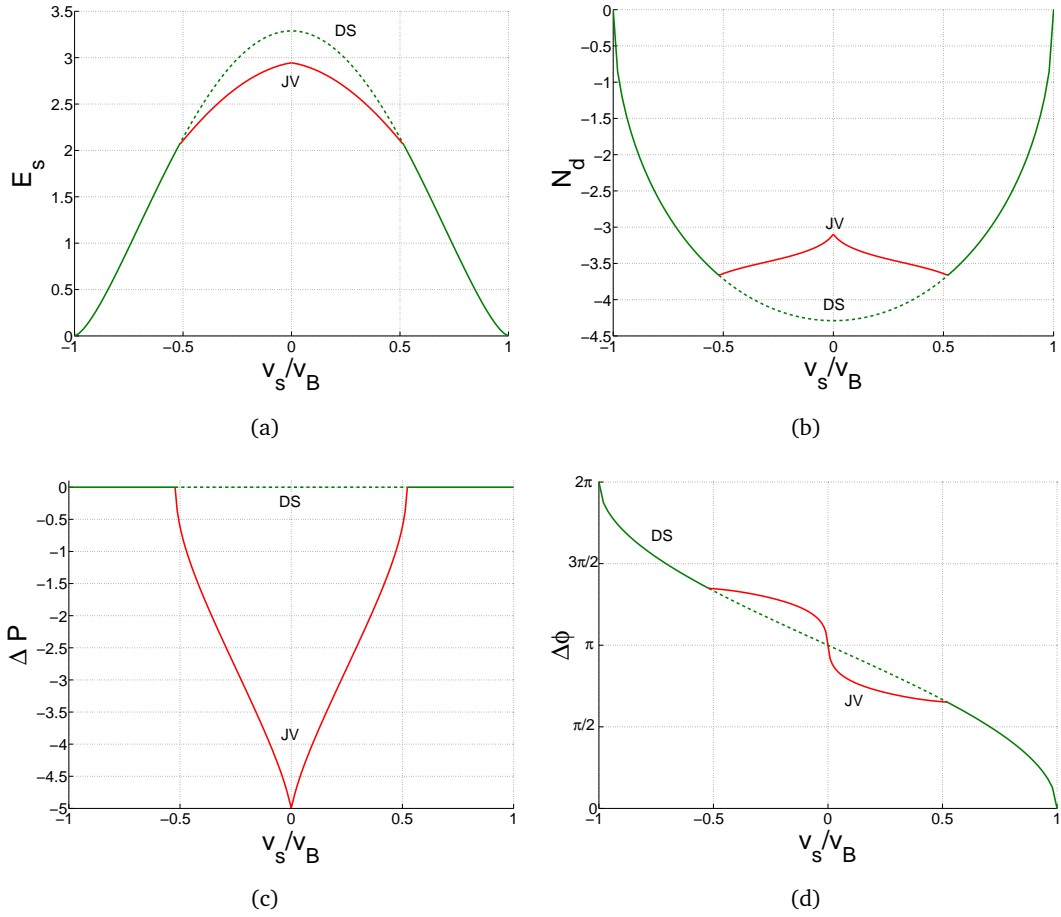

Figure 6: No cross-interaction, intermediate coupling regime ($\gamma = 1$, $\nu = 0.15$). Energy (a), missing particle number (b), angular momentum (c) and phase difference (d) as a function of velocity. Properties of Josephson vortices (labeled 'JV') are plotted in red and those of dark solitons (labeled 'DS') in green. Solid lines indicate stable solutions, while dashed lines depict unstable ones.

vortices are stable and dark solitons are unstable and when Josephson vortices cease to exist, dark solitons become stable. This fact was exploited in Ref. [44], where the authors present a similar plot to Fig. 9 (a) based on a stability calculation for dark solitons. An outline of the stability calculation is presented in Appendix A.

When $\gamma < 1$ we see that once again there exists a critical momentum beyond which the Josephson vortex solutions do not exist, but the Josephson vortex dispersion relation now terminates by touching the dark soliton dispersion relation non-tangentially (*i.e.* the slopes of the curves are different). The critical velocity is plotted as a function of $\gamma$ in Fig. 9 (b). The staggered soliton branch terminates at the exact same critical momentum and velocity as the Josephson vortex branch.

In the $\gamma < 1$ regime, Josephson vortices are again always stable, but the situation for dark solitons is quite different. Figure 10 shows a numerically-determined boundary line (plotted with blue circles) in the $P_c$-$\gamma$ plane such that above this curve, dark solitons are unstable and below it they are stable. As soon as dark solitons become stable, staggered solitons appear. These are always unstable except exactly at $\gamma = 0$ (the entire Manakov family of solutions is always stable). There exist small regions of stability in Fig. 10, bounded by the almost vertical sections of the stability-flip curve and $0, 4\pi(1 + \nu)$, the limits of $P_c$. These are regions where

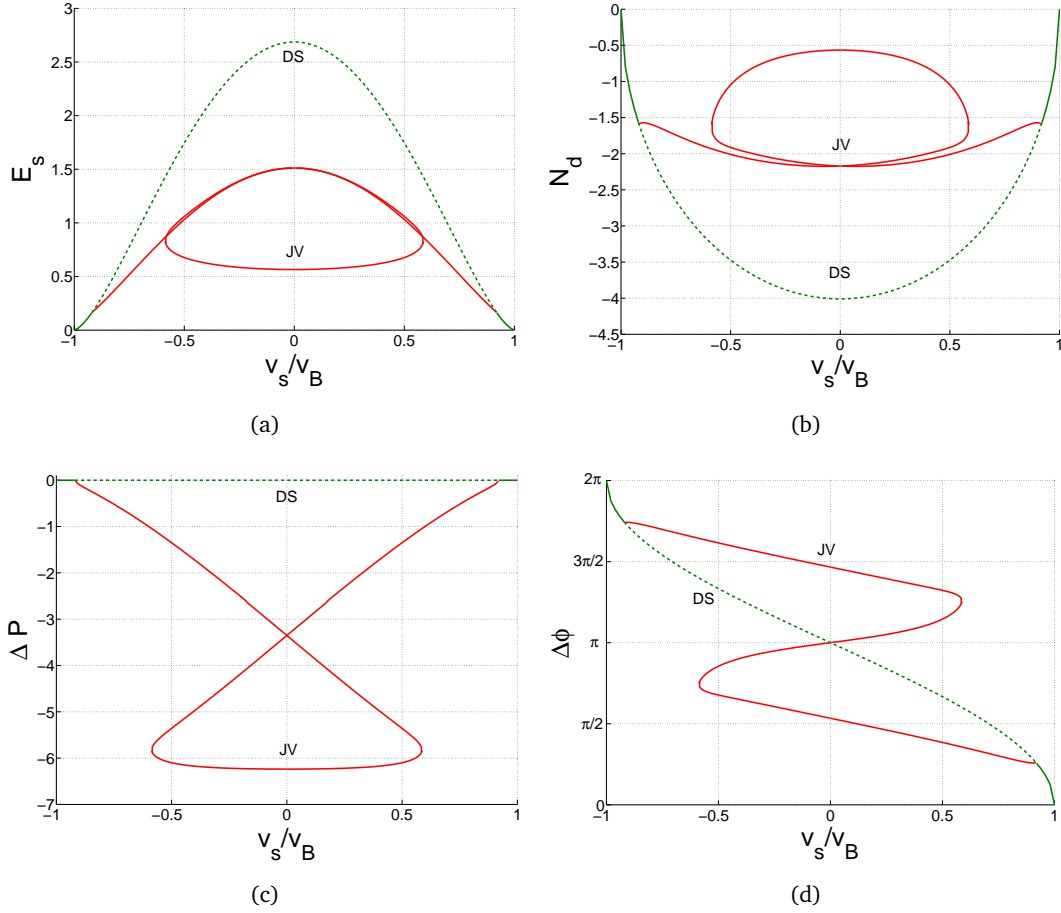

Figure 7: No cross-interaction, weak coupling regime ($\gamma = 1$, $\nu = 0.005$). Energy (a), missing particle number (b), angular momentum (c) and phase difference (d) as a function of velocity. Properties of Josephson vortices (labeled 'JV') are plotted in red and those of dark solitons (labeled 'DS') in green. Solid lines indicate stable solutions, while dashed lines depict unstable ones.

staggered solitons and Josephson vortices do not exist and dark solitons are stable (as in the regime $\gamma = 1$). The development of these slivers of stability is seen in Fig. 9 (b) as a dip of the critical velocity, starting at about $\gamma = 0.86$.

For zero velocity dark solitons, we can analytically compute the points in parameter space where stability changes – this is done in Appendix A.1. For $\nu = 0.005$ as in Fig. 10, the result is $\gamma = 0.975$, in agreement with numerical calculations (this point has been added to Fig. 10 as a red square). In fact, the analytical calculation also allows one to see that this stability-flip point starts at $\gamma = 1$ when $\nu = 0$, smoothly decreases and reaches $\gamma = 0$ at $\nu = 1/3$, so that outside of $0 < \nu < 1/3$, neither Josephson vortices nor staggered solitons exist.

Thus, there is a region of bistability for $\gamma < 1$ where Josephson vortices (lowest energy) and dark solitons (highest energy) are both stable with the unstable staggered soliton branch (intermediate energy) between them. An illustration is given in Fig. 11 where we fix $\nu = 0.005$, $\nu_s = 0$, $P_c = 2\pi(1 + \nu)$ and plot the energy as a function of $\gamma$. The energies of dark solitons and Josephson vortices are constant since the solutions (16) and (21) are independent of $\gamma$, as is the energy functional (9) when $|\psi_1|^2 = |\psi_2|^2$. Overall, this has the familiar shape of a bistability bifurcation diagram with a fold. The unusual features are that the upper branch continues to the right past the fold and that the three lines do not make a single, smooth curve.

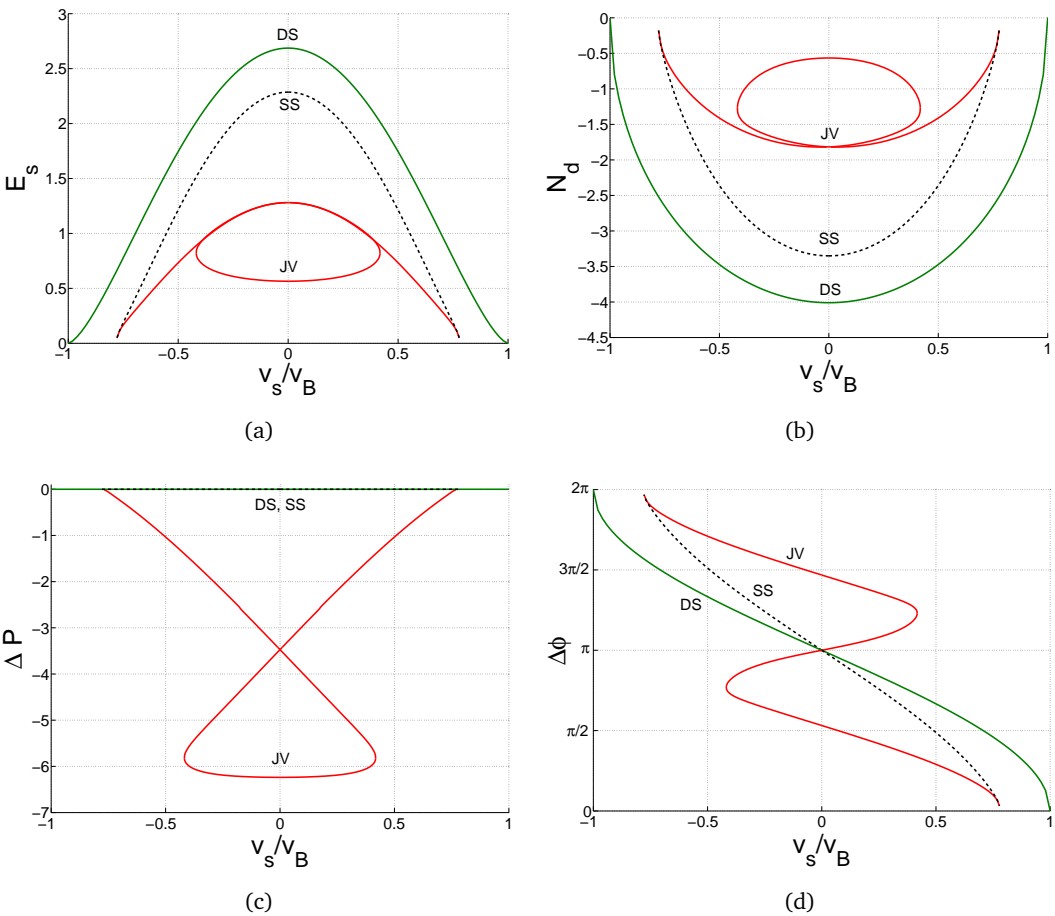

Figure 8: Intermediate cross-interaction, weak coupling regime ($\gamma = 0.5$, $\nu = 0.005$). Energy (a), missing particle number (b), angular momentum (c) and phase difference (d) as a function of velocity. Properties of Josephson vortices (labeled 'JV') are plotted in red, those of dark solitons (labeled 'DS') in green, and quantities associated with staggered solitons (labeled 'SS'), in black. Solid lines indicate stable solutions, while dashed lines depict unstable ones.

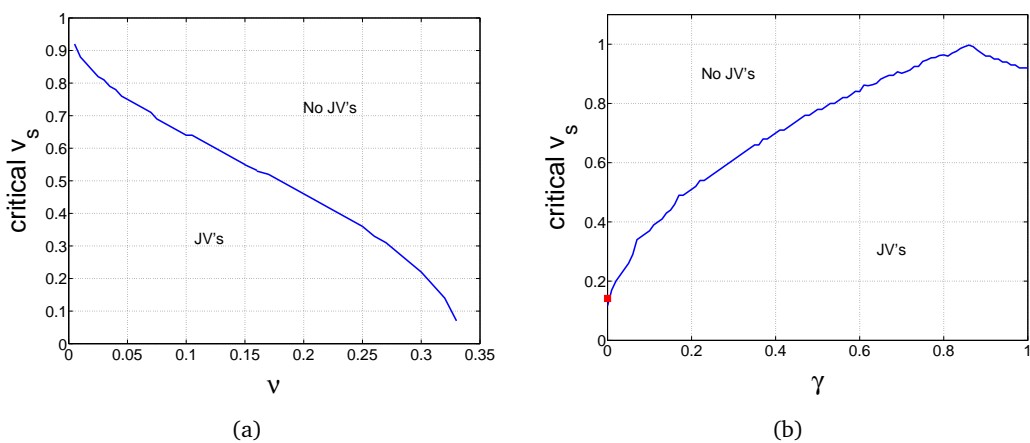

Figure 9: Critical velocity (maximum allowed speed) for Josephson vortices as a function of $\nu$ with $\gamma = 1$ (a) and as a function of $\gamma$ with $\nu = 0.005$ (b). The red square is the analytical Manakov result and the full (blue) lines are numerical results.

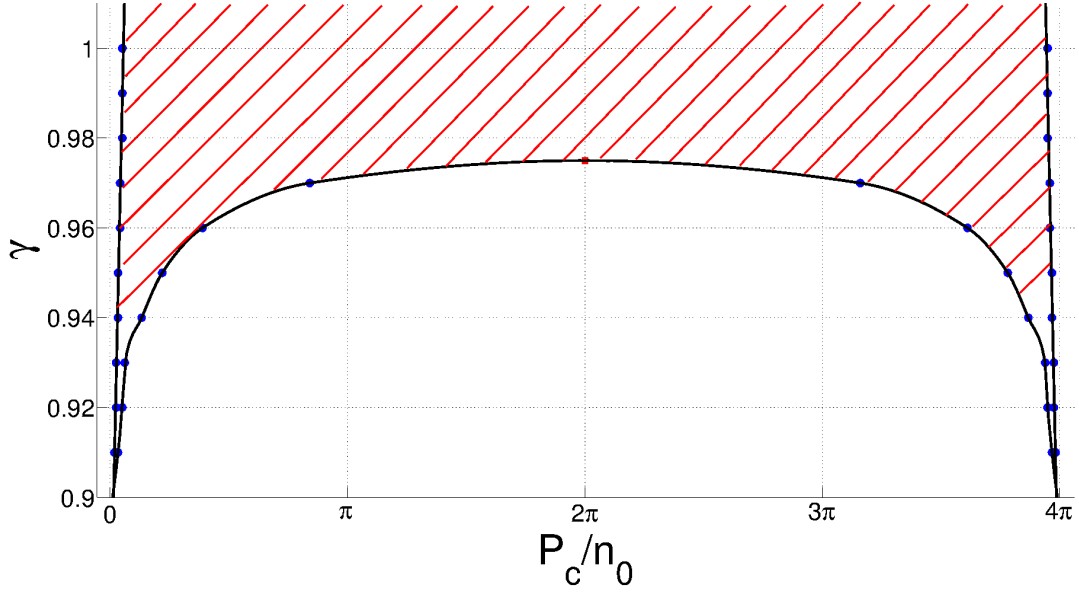

Figure 10: Stability diagram for dark (grey) solitons at $v = 0.005$. Dark solitons are unstable in the shaded region. The boundary points are calculated analytically (red square, see Appendix A.1) and numerically (blue dots) and the black line is a guide to the eye (spline fit to the data points). In the stable region below the black line unstable staggered solitons coexist with dark solitons. In the stability region at small and large $P_c$ all excitation branches have merged and only stable grey solitons exist.

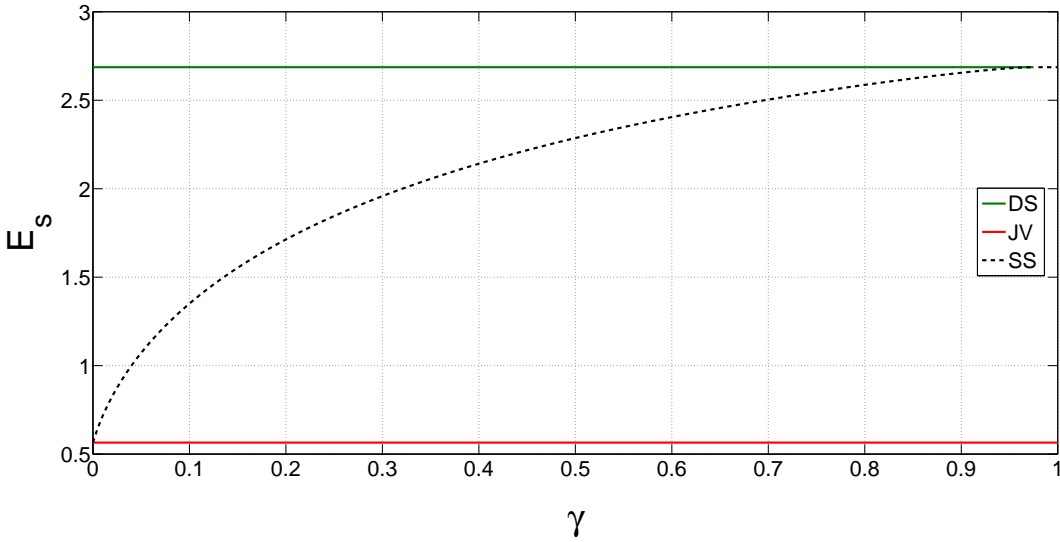

Figure 11: Excitation energy of stationary (unstable) staggered solitons (labeled 'SS') as a function of $\gamma$ at $v = 0.005, P_c = 2\pi n_0$ (black dashed line). The energy of the stable dark soliton (top green line, labeled 'DS') and Josephson vortex (lower red line, labeled 'JV') are also shown.

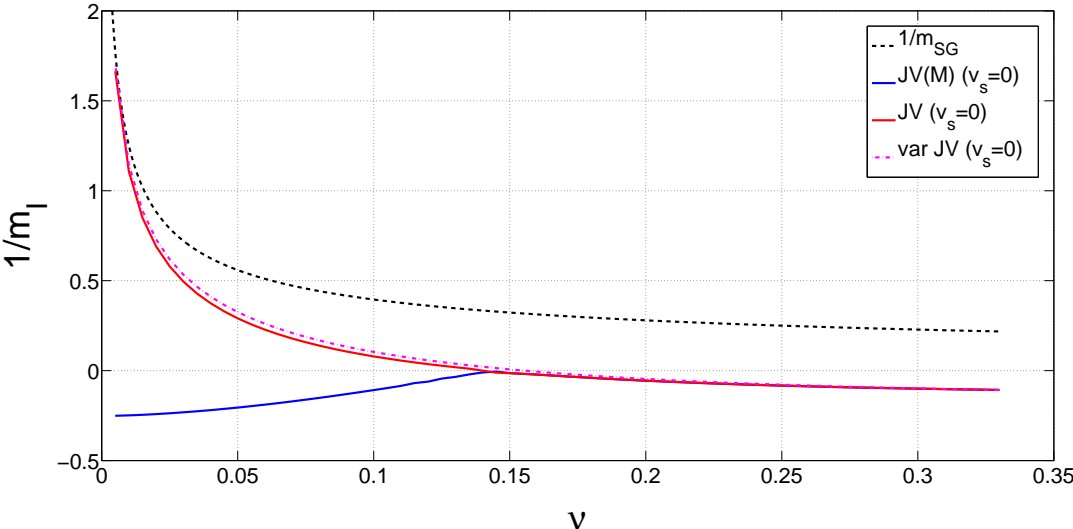

Figure 12: Inverse of the inertial mass for Josephson vortices (upper red full line, labeled 'JV') and Josephson vortex maxima (lower blue full line, labeled 'JV(M)') at $v_s = 0$ as a function of tunneling strength with $\gamma = 1$ obtained from numerical solutions. The magenta dash-dotted line is an approximate result obtained from a variational calculation for Josephson vortices (labeled 'var JV'), equation (33). The black dashed line shows $1/m_{SG}$ from (47), discussed is section 12.

## 9  Inertial mass and missing particle number

In this section we focus on the first parameter range ($\gamma = 1$) and examine some overall properties of the dispersion relations. To start with, we can calculate the inertial mass of Josephson vortices and Josephson vortex maxima (evaluating the derivatives in (13) at the minimum and maximum of the dispersion relation, respectively) as a function of $v$, which yields Fig. 12. The blue and red solid curves were obtained from the numerical Josephson vortex solutions. We define the bifurcation point at which the central part of the Josephson vortex dispersion relation changes concavity by the $v$ value at which the $1/m_I$ curve (red solid line in Fig. 12) crosses zero. This happens at $v = 0.1413$. The magenta dash-dotted line shows the variational approximation for Josephson vortices (see section 10). The black dashed line will be described in section 12.

The inertial mass is a useful characteristic of an excitation, but the experimentally-accessible quantity is $m_I/N_d$, the ratio of the inertial mass to the number of particles in the excitation, as it relates to the experimentally-measurable frequency ratio of small amplitude oscillations in the presence of weak harmonic trapping [9, 10, 49, 50]. With this in mind, Fig. 13 shows $N_d$ at the extrema of the Josephson vortex dispersion relation as a function of $v$, and Fig. 14 shows the ratio $N_d/m_I$ obtained by combining the data from Figs. 12 and 13. The magenta dash-dotted line shows the variational approximation for Josephson vortices, described in section 10. It is clear that the red curve certainly crosses zero, which means that $m_I/N_d \to \pm\infty$ on either side of the critical point. This implies that essentially, the Josephson vortices become infinitely heavy.

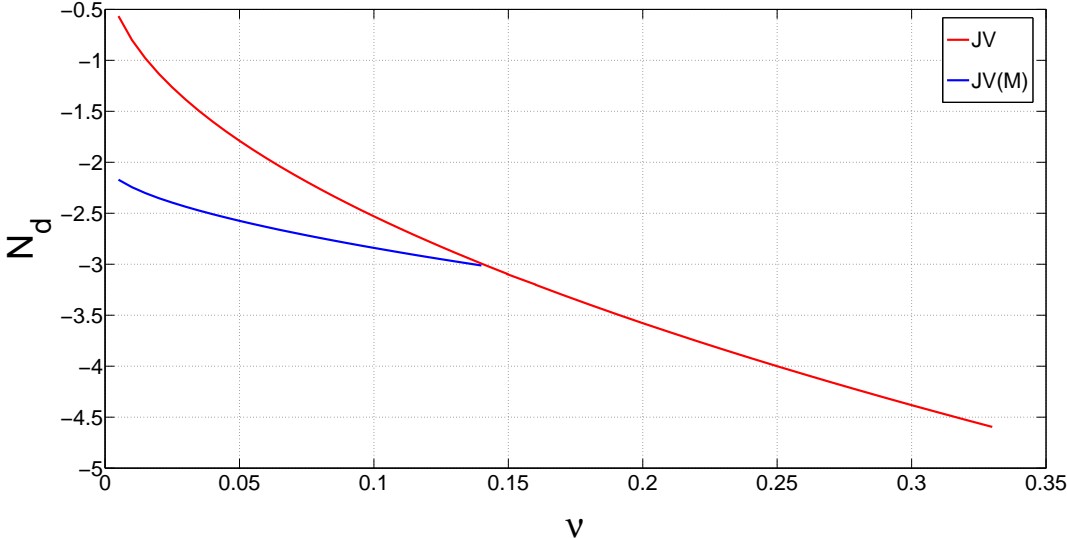

Figure 13: Missing particle number for Josephson vortices (red upper curve, labeled 'JV') and Josephson vortex maxima (blue lower curve, labeled 'JV(M)') as a function of tunneling strength with $\gamma = 1$, evaluated at the extrema of the dispersion relation (*i.e.* $v_s = 0$).

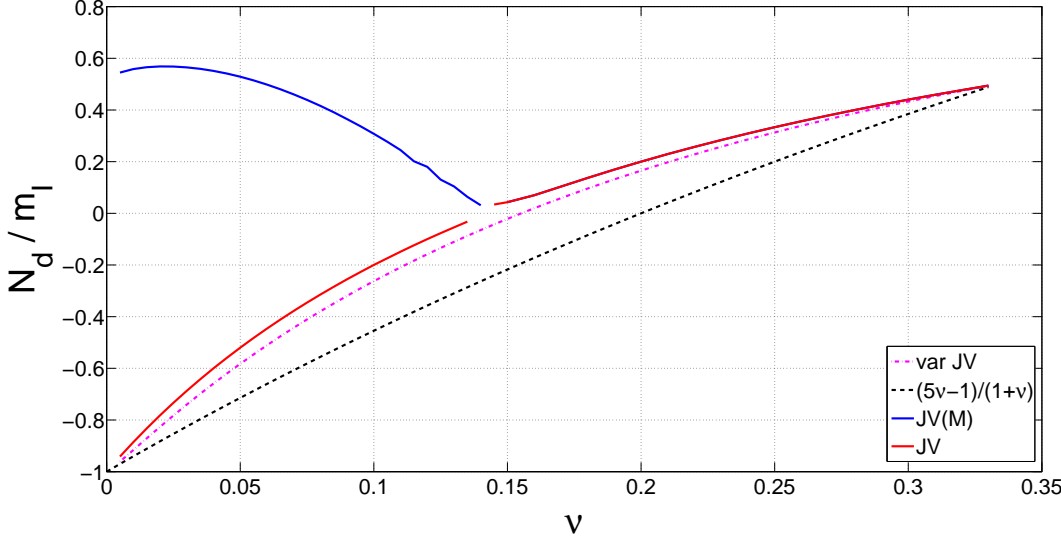

Figure 14: Missing particle number over inertial mass for Josephson vortices (red lower solid line, labeled 'JV') and Josephson vortex maxima (blue upper solid line, labeled 'JV(M)') as a function of tunneling strength with $\gamma = 1$, evaluated at the extrema of the dispersion relation (*i.e.* $v_s = 0$). The magenta dash-dotted line is an approximate result obtained from a variational calculation for Josephson vortices (labeled 'var JV'), and the black dashed line is a prediction from Ref. [44].

## 10 Variational calculation for Josephson vortices

In light of the results of the previous section, we endevour to find a variational approximation for Josephson vortices near $v_s = 0$, $P_c = 2\pi(1 + v)$, *i.e.* in the immediate vicinity of the known analytical solution (21). We take the variational ansatz

$$\psi_{1,2} = \sqrt{1 + v}\left\{ i\sin(\alpha) + \cos(\alpha)\tanh(Az) \pm iB_{1,2}\operatorname{sech}(Az)e^{iz\varepsilon} \right\}, \qquad (28)$$

a form general enough to capture dark solitons, zero-velocity Josephson vortices and Manakov solitons. One then has to evaluate $\mathscr{L} = E - v_s P_c$ for this variational guess and take away $\mathscr{L}$ for the background state, resulting in the difference, $\Delta\mathscr{L}$. Differentiating $\Delta\mathscr{L}$ with respect to all five variational parameters $(A, \alpha, B_1, B_2, \varepsilon)$ and setting the resulting expressions to zero, we obtain a system of five coupled non-linear equations. These are quite complicated, and a direct solution is impractical. Instead, we linearize the equations in $v_s$: we set $A = A_0 + v_s\tilde{A}$, $\varepsilon = \varepsilon_0 + v_s\tilde{\varepsilon}$, $\alpha = \alpha_0 + v_s\tilde{\alpha}$, $B_{1,2} = B_0 + v_s\tilde{B}_{1,2}$, where the zeroth order parameters are chosen to correspond with the solution (21): $A_0 = 2\sqrt{\nu}$, $B_0 = \sqrt{\frac{1-3\nu}{1+\nu}}$, $\varepsilon_0 = \alpha_0 = 0$. The zeroth-order terms in the linearized equations thus cancel, and it remains to set the first order terms (in $v_s$) to zero. Introducing $\tilde{B}_\pm = \tilde{B}_1 \pm \tilde{B}_2$, we replace the equations resulting from $d\Delta\mathscr{L}/dB_{1,2} = 0$ by the sum and difference of these two equations. The five equations we must now solve decouple into two sets: two- and three-coupled equations. The solutions are: $\tilde{A} = \tilde{B}_+ = 0$, and

$$
\begin{aligned}
\Omega = &-48\left\{-\gamma^2 + 2(\gamma-2)\gamma\nu + \nu^2\left[24 + \gamma(44+3\gamma)\right]\right\}\left[3\nu + \gamma(6\nu-2)\right] \\
&-4\pi^2\left[2\nu + \gamma(3\nu-1)\right]\left[3\gamma\nu(7-29\nu) - 54\nu^2 + 5\gamma^2(1+\nu)(3\nu-1)\right] \\
&+3\gamma\pi^4(1+\nu)(\gamma-2\nu-3\gamma\nu)^2,
\end{aligned}
\tag{29}
$$

$$
\begin{aligned}
\tilde{\alpha} = \sqrt{\nu}\Big\{&216\nu^2(\pi^2-8) + 6\gamma\nu\left[168 - 888\nu + 4\pi^2(19\nu-5) + \pi^4(1+\nu)\right] \\
&+\gamma^2(3\nu-1)\left[-96(1+13\nu) + 4\pi^2(13\nu-5) + 3(1+\nu)\pi^4\right]\Big\}/\Omega,
\end{aligned}
\tag{30}
$$

$$
\tilde{\varepsilon} = 72(1+2\gamma)\nu^2\left[6\nu(\pi^2-8) + \gamma(3\nu-1)(3\pi^2-32)\right]/\Omega,
\tag{31}
$$

$$
\tilde{B}_- = 144\gamma(1+2\gamma)\nu^{3/2}\pi(3\nu-1)/\Omega.
\tag{32}
$$

Linearising the variational equations in $v_s$ is an approximation that is of the same order as keeping terms up to $\mathcal{O}(v_s^2)$ in $E_s$ (the excitation energy) and $\mathcal{O}(v_s)$ in $P_c$ (the total momentum). Making such an expansion we can calculate the inertial mass $m_I = 2\frac{dE_s}{d(v_s^2)}$, to obtain

$$
\begin{aligned}
m_I = \frac{8\sqrt{\nu}}{\Omega}\Big\{&48\left[\gamma + 2\nu(3+\gamma) + \nu^2(30+49\gamma)\right]\left[3\nu + \gamma(6\nu-2)\right] \\
&-3\gamma(1+\nu)^2\pi^4(2\nu + \gamma\pi^4(3\nu-1)) \\
&-4\pi^2\left[27\nu^2(1+5\nu) + 3\gamma\nu(\nu(137\nu-14)-7) + \gamma^2(5+\nu(\nu(309\nu-133)-5))\right]\Big\},
\end{aligned}
\tag{33}
$$

which is plotted in Fig. 12 alongside the numerical results. Using the zero-velocity solution (21), we can compute the missing particle number at $v_s = 0$ as $N_d = -8\sqrt{\nu}$. The ratio $N_d/m_I$ from this calculation is shown in Fig. 14 as the magenta dash-dotted line. Note that Ref. [44] predicted $N_d/m_I = (5\nu-1)/(1+\nu)$, which is also displayed in Fig. 14 for comparison.

## 11 The sine-Gordon equation

The second parameter regime that we have investigated ($\nu = 0.005$) is particularly interesting in terms of how it compares to the analytically-solvable sine-Gordon model. In order to carry out such a comparison, we first give a brief review of the sine-Gordon equation.

In Appendix B we derive the sine-Gordon equation from the model of section 2 by assuming that the densities of the two fields are practically equal to each other and are almost constant. In addition, we isolate the terms from the Lagrangian density that contribute to the relative phase sector, which asymptotically decouples from the total phase sector in the limit of vanishing tunneling. While the total phase sector supports gapless elementary excitations,

it is the relative phase sector that is captured by the sine-Gordon model and is relevant for the Josephson vortices. This selection of terms is partly justified *a posteriori* by the success of the analysis we perform in section 12.

The derivation of Appendix B allows one to express the sine-Gordon parameters through the Gross-Pitaevskii model parameters, thus enabling a direct comparison of the two models. In this section we will present some analytical results for the sine-Gordon equation [57], written with parameters determined by the procedure in Appendix B.

The Lagrangian density of the sine-Gordon model is

$$\mathcal{L} = \frac{\hbar^2}{4(g-g_c)}(\partial_t \phi_a)^2 - \frac{\hbar^2}{4m}\frac{\mu+J}{g+g_c}(\partial_x \phi_a)^2 + 2J\frac{\mu+J}{g+g_c}\cos(\phi_a), \tag{34}$$

where

$$\phi_a = \phi_1 - \phi_2. \tag{35}$$

The Hamiltonian density can be obtained in the usual way:

$$\begin{aligned} P_\phi &= \frac{\partial \mathcal{L}}{\partial(\partial_t \phi_a)}, \\ \mathcal{H} &= P_\phi(\partial_t \phi_a) - \mathcal{L}, \end{aligned} \tag{36}$$

where $P_\phi$ is the canonical conjugate coordinate to $\phi_a$. The Euler-Lagrange equation

$$\frac{\partial \mathcal{L}}{\partial \phi_a} - \partial_x \frac{\partial \mathcal{L}}{\partial(\partial_x \phi_a)} - \partial_t \frac{\partial \mathcal{L}}{\partial(\partial_t \phi_a)} = 0 \tag{37}$$

yields the sine-Gordon equation:

$$\partial_{tt}\phi_a - \frac{\gamma}{m}(\mu+J)\partial_{xx}\phi_a = -\frac{4J\gamma(\mu+J)}{\hbar^2}\sin(\phi_a). \tag{38}$$

Rewriting in dimensionless form (see (5)) and in a frame moving at $v_s$, the sine-Gordon equation becomes

$$\left[v_s^2 - \gamma(1+\nu)\right]\partial_{zz}\phi_a + 4\nu\gamma(1+\nu)\sin(\phi_a) = 0. \tag{39}$$

The solution is given by

$$\begin{aligned} \zeta &= \sqrt{\frac{4\nu\gamma(1+\nu)}{\gamma(1+\nu)-v_s^2}}, \\ \phi_a &= 4\tan^{-1}\left(e^{\zeta z}\right). \end{aligned} \tag{40}$$

The Hamiltonian density is

$$\mathcal{H} = \frac{1}{4}\left[\frac{v_s^2}{\gamma}+1+\nu\right](\partial_z \phi_a)^2 - 2\nu(1+\nu)\cos(\phi_a), \tag{41}$$

and the excitation energy is

$$E_s = \frac{8\nu(1+\nu)}{\zeta} + 2\zeta\left(1+\nu+\frac{v_s^2}{\gamma}\right). \tag{42}$$

Next, using

$$P_c(v_s) = \int_0^{v_s} d\bar{v}_s \frac{1}{\bar{v}_s}\frac{dE_s}{d\bar{v}_s}, \tag{43}$$

we get the canonical momentum as

$$P_c = \frac{4v_s}{\gamma}\zeta. \tag{44}$$

We can eliminate $v_s$ to get the dispersion relation:

$$E_s^2 = (1+v)\left[\gamma P_c^2 + 64v(1+v)\right], \tag{45}$$

or if we choose to write (in analogy to a relativistic particle)

$$E_s^2 = m_{SG}^2 c_{SG}^4 + c_{SG}^2 P_c^2, \tag{46}$$

then we identify

$$\begin{aligned} m_{SG} &= \frac{8\sqrt{v}}{\gamma}, \\ c_{SG} &= \sqrt{\gamma(1+v)}, \end{aligned} \tag{47}$$

as the "mass" and "speed of light" of the sine-Gordon soliton, respectively.

## 12    Relativistic behavior

We have seen that at $\gamma = 1$ and small $v$, the coupled-BECs Josephson vortex dispersion relation develops a dip about $P_c = 2\pi(1+v)$ (see Fig. 5), similar in shape to the central part of the dispersion relation of the sine-Gordon equation. The equivalence of the two models in this regime has been suggested before [30], and now that we have the sine-Gordon dispersion relation expressed through the Gross-Pitaevskii model parameters, we are in a position to check this statement.

First, we can compare the dispersion relations visually. This is shown in Fig. 15, and the Josephson vortex dispersion relation indeed seems to be very close to the sine-Gordon curve near the zero-velocity point $P_c = 2\pi n_0$. Next, we would like to compare the sine-Gordon parameters $m_{SG}$ and $c_{SG}$ to their equivalents in the coupled BECs model as a function of $v$. A sensible way of extracting these parameters from the Josephson vortex dispersion relation is to first obtain $c_{JV}$ from

$$c_{JV} = \sqrt{\max\left(\frac{dE_s^2}{dP_c^2}\right)}, \tag{48}$$

using data about $P_c = 2\pi(1+v)$, and then obtain $m_{JV}$ as

$$m_{JV} = \sqrt{\frac{E_s^2(P_c = 2\pi(1+v))}{c_{JV}^4}}. \tag{49}$$

The "relativistic mass" $m_{JV}$ calculated this way (for $v \leq 0.14$) is indistinguishable from $m_I$ obtained as a derivative using equation (13) plotted in Fig. 12 as a red solid line. Comparing the red line to the black dashed line ($m_{SG}$) in Fig. 12, it appears that the Josephson vortex mass $m_{JV}$ indeed approaches the sine-Gordon result as $v \to 0$. Note that we are unable to compute numerical Josephson vortex solutions at smaller $v$ because the excitation length-scale becomes unmanageable.

As for the "speed of light", $c_{JV}$, Fig. 16 shows that the functional dependence on $v$ is completely different for the coupled BECs and sine-Gordon models, and it is clear that the two only become equal at $v = 0$ but the slopes remain different. We therefore conclude that the Gross-Pitaevskii model approaches the sine-Gordon model only asymptotically.

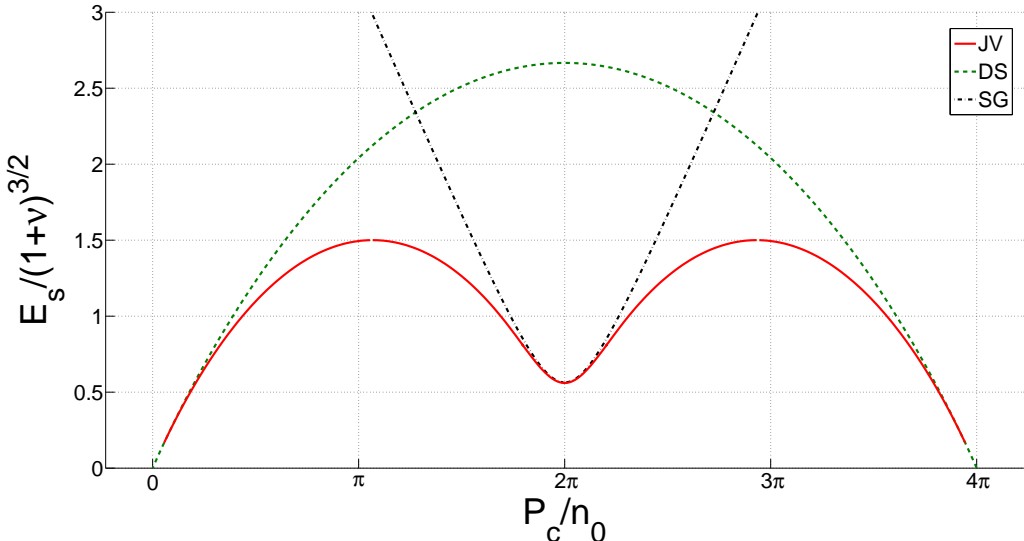

Figure 15: Dispersion relation of the coupled BECs model and the sine-Gordon equation with $\gamma = 1$, $\nu = 0.005$. Green dashed line – dark solitons (labeled 'DS'), red solid line – Josephson vortices (labeled 'JV'), black dash-dotted line – sine-Gordon solutions (labeled 'SG'). The Josephson vortex dispersion relation is very close to the sine-Gordon dispersion relation about $P_c = 2\pi(1 + \nu)$ (note that this latter curve was shifted horizontally to $P_c = 2\pi(1 + \nu)$).

There are two fundamental speeds in the coupled-BECs model, which can be found by computing linearized excitations about the vacuum state, as was done in [6]. The authors find two elementary excitation branches: gapless Bogoliubov phonons (subscript "B") and a gapped relative-phase excitations (subscript "RP"). A standard Bogoliubov calculation (such as the one in Appendix A) leads to the dimensionless oscillation frequencies

$$\omega_B = \sqrt{1 + \nu}\sqrt{\frac{1}{2}k^2\left(\frac{1}{2(1 + \nu)}k^2 + 2\right)}, \tag{50}$$

$$\omega_{RP} = \sqrt{\left(\frac{1}{2}k^2 + 2\nu\right)\left(\frac{1}{2}k^2 + 2\gamma(1 + \nu) + 2\nu\right)}, \tag{51}$$

where $k$ is a dimensionless wavenumber. If for some sufficiently small $k$ the frequency $\omega$ becomes imaginary, the vaccum state is unstable. Thus, the vacuum can become unstable if $\gamma < 0$. The speeds associated with each branch are the speed of sound, $c_B = \sqrt{1 + \nu}$, and $c_{RP} = \sqrt{\gamma(1 + \nu) + 2\nu}$, which can be interpreted as a "speed of light". Both the elementary speeds are shown in Figs. 16 and 17 for comparison with sine-Gordon and Josephson vortex results. Notice that $c_{RP}$ is never equal to (the variational) $c_{JV}$ for $\gamma > 0$.

Figure 17 finally explores the regime of finite cross-nonlinearity, $\gamma < 1$. Here we compare the sine-Gordon "speed of light" $c_{SG}$ to its equivalent from the Gross-Pitaevskii model $c_{JV}$ (showing both a numerical calculation and a variational approximation), and to the elementary speeds $c_B$ and $c_{RP}$. We can see that the difference between $c_{SG}$ and $c_{JV}$ remains constant as a function of $\gamma$ (it only depends on $\nu$) and that both the Josephson vortex and sine-Gordon "speeds of light" exhibit a square-root dependence on $\gamma$ (recall that $c_{SG} = \sqrt{\gamma(1 + \nu)}$) while $c_B$ is independent of $\gamma$. Thus, by decreasing $\gamma$ at a small $\nu$ we can decouple two fundamental speeds in the Gross-Pitaevskii model.

We can also use the results of section 10 to obtain an approximate analytical expression for the "speed of light". The excitation energy of the stationary vortex (21) is $E_s = \frac{8}{3}(3 - \nu)\sqrt{\nu}$, which together with $m_I$ of equation (33), yields the "speed of light" as $c = \sqrt{\frac{E_s}{m_I}}$, which is

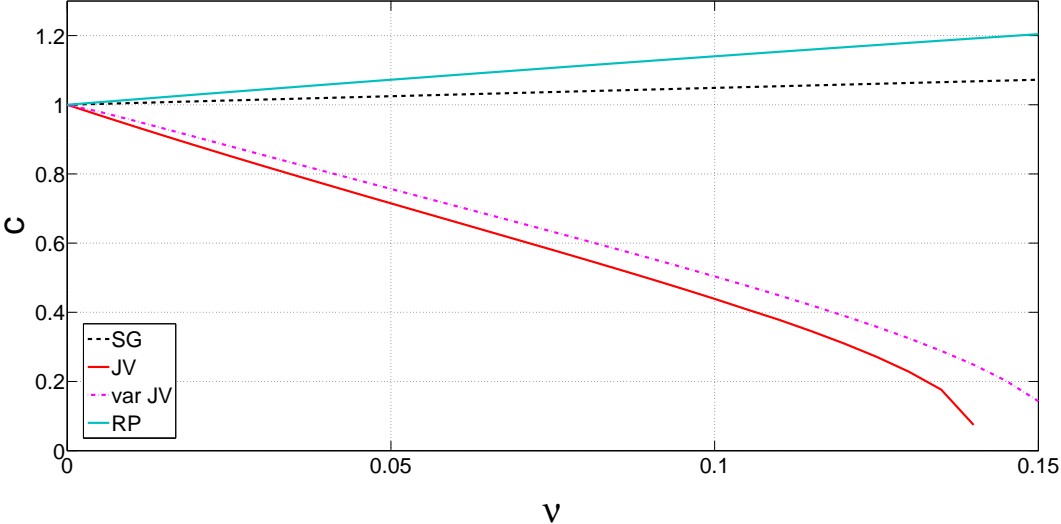

Figure 16: "Speed of light" from a relativistic dispersion relation for the sine-Gordon (labeled 'SG') and coupled BECs models (labeled 'JV') at $\gamma = 1$. The magenta dash-dotted line is an approximate result obtained from a variational calculation for Josephson vortices (labeled 'var JV'). Note that for $\gamma = 1$, $c_{SG} = c_B$, the speed of sound, and the elementary "speed of light" $c_{RP}$ is added as a solid cyan line (upper).

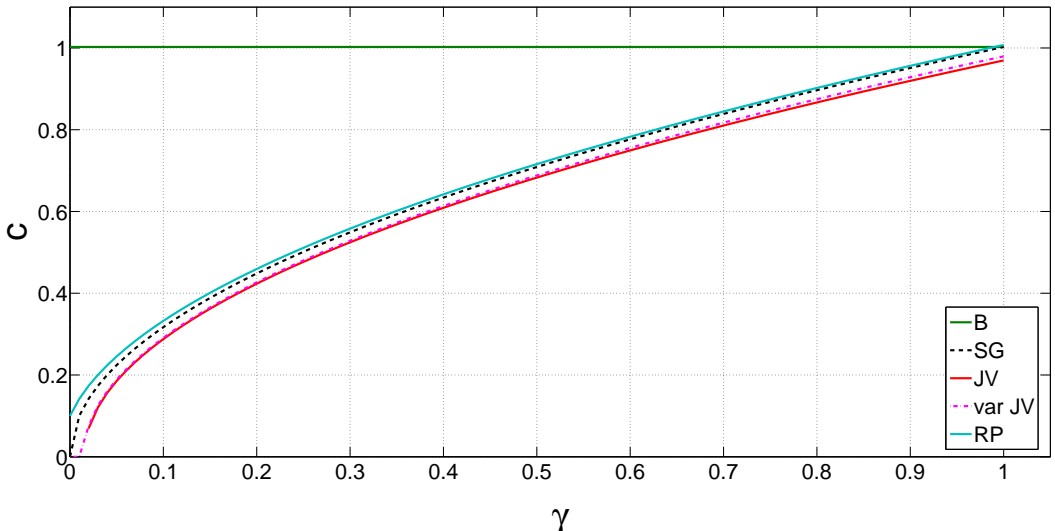

Figure 17: A comparison of the numerical $c_{JV}$ from the Gross-Pitaevskii model (red lower solid line, labeled 'JV'), the variational $c_{JV}$ (dash-dotted magenta line, labeled 'var JV'), $c_{SG}$ of the sine-Gordon equation (black dashed line, labeled 'SG'), the speed of sound $c_B$ (green upper solid line, labeled 'B') and the elementary "speed of light" $c_{RP}$ (cyan intermediate solid line, labeled 'RP'). For all curves, $v = 0.005$.

added to Figs. 16 and 17.

# 13   Discussion and conclusions

We have carried out numerical and analytical investigations of the solitary wave solutions of a model of two linear, parallel, long coupled BECs. The model has three distinct parameters: $\nu$ (representing coupling between the condensates), $\gamma$ (which carries information about self- and cross- non-linearities of the fields), and $v_s$ (the uniform translation speed of localized excitations). This model has three types of solutions: dark solitons, Josephson vortices and a new set of solutions which we have labeled staggered solitons. Analytical expressions are available for dark solitons (for arbitrary parameters), zero-velocity Josephson vortices (but not Josephson vortex maxima), and the Manakov solutions for $\gamma = 0$. Numerically we have found the full dispersion relations for all solutions in the parameter regimes $\gamma = 1$, $0 < \nu < 1/3$ and $\nu = 0.005$, $0 \leq \gamma \leq 1$.

In the absence of cross-nonlinearity ($\gamma = 1$), there is a critical point at $\nu \approx 0.1413$ where the Josephson vortex dispersion relation at $P_c = 2\pi(1+\nu)$ changed concavity. This corresponds to the inertial mass changing sign, going through $\pm\infty$. Thus, very "heavy" Josephson vortices can be created by tuning the coupling strength in this range. The heavy solitonic vortices observed experimentally in [9,10] are closely related, but there it is not possible to change the sign of $m_I$ by tuning a parameter.

The coupled BECs Josephson vortex dispersion relation at small $\nu$ and $\gamma = 1$ can be compared to the dispersion relation of the integrable sine-Gordon equation. The sine-Gordon parameters may be expressed through the Gross-Pitaevskii model parameters by deriving the sine-Gordon model from the Gross-Pitaevskii model in the small $\nu$ limit. We found that the Josephson vortex dispersion relation about $P_c = 2\pi(1 + \nu)$ became equivalent to the sine-Gordon one asymptotically for $\nu \to 0$ but that the characteristic velocity scales of the Josephson vortices and the gapped linear excitations differ for finite $\nu$. This challenges the widely-used approximation, or at least suggests some caution in its application. However, by working in the small $\nu$ regime, Josephson vortices may open the possibility for experimental study of "relativistic particles" (to a good approximation) using collective excitations of ultra-cold atoms.

When $\gamma < 1$, there exists a $\gamma$- and $\nu$- dependent region where dark solitons and Josephson vortices are both stable, separated (in energy) by the unstable staggered solitons. For $\gamma = 1$, dark solitons are always unstable and Josephson vortices are stable. Therefore, observing dark solitons in the latter regime could be difficult because they would quickly decay into two opposite-circulation Josephson vortices. If one worked in the bistable region, however, since dark solitons are dynamically stable they would not decay. This could potentially enable one to observe the dynamics and interaction of Josephson vortices with dark solitons experimentally. Note, however, that in the bistable regime dark solitons are still thermodynamically unstable, and thus the presence of a thermal cloud in a cold-atom experimental setting would lead to the eventual decay of the dark soliton, due to its negative inertial mass [1,58]. Josephson vortices around the minimum of the dispersion relation, however, should be stabilized at finite temperature.

## Acknowledgements

We would like to thank Dr. Oleksandr Fialko for useful discussions.

**Funding information**   This work was partially supported by the Marsden fund of New Zealand (Grant No. MAU1604). SS acknowledges support from the Massey University Doctoral Research Dissemination Grant during the publication process.

# A  Stability calculation

This appendix gives details of how the stability of a solution to equations (6) is determined. We start from the Gross-Pitaevskii equations in dimensionless form, allowing for additional time dependence, other than mere translation at $v_s$:

$$i\partial_\tau \psi_k = -\frac{1}{2}\partial_{zz}\psi_k + iv_s\partial_z\psi_k - \psi_k + \frac{1}{2}(1+\gamma)|\psi_k|^2\psi_k + \frac{1}{2}(1-\gamma)|\psi_{3-k}|^2\psi_k - \nu\psi_{3-k}. \quad (52)$$

To find out whether a solution is stable we must add a variation to the wavefunction:

$$\psi_k(z) \to \psi_k(z) + \delta\psi_k(z,\tau). \quad (53)$$

The right-hand side of (53) is substituted into (52); zeroth-order terms in $\delta\psi_k$ give the unperturbed equations (52), terms of second order in $\delta\psi_k$ and higher are discarded, and the first order terms give two linear equations for $\delta\psi_k$:

$$\begin{aligned}
i\partial_\tau \delta\psi_k = &-\frac{1}{2}\partial_{zz}\delta\psi_k + iv_s\partial_z\delta\psi_k - \delta\psi_k - \nu\delta\psi_{3-k} \\
&+ \frac{1}{2}(1+\gamma)\big[2|\psi_k|^2\delta\psi_k + \psi_k^2\delta\psi_k^*\big] + \frac{1}{2}(1-\gamma)\times \\
&\big[|\psi_{3-k}|^2\delta\psi_k + \psi_{3-k}\psi_k\delta\psi_{3-k}^* + \psi_{3-k}^*\psi_k\delta\psi_{3-k}\big].
\end{aligned} \quad (54)$$

We then make the ansatz

$$\delta\psi_k(z,\tau) = a_k(z)e^{-i\lambda\tau} + b_k^*(z)e^{i\lambda^*\tau}. \quad (55)$$

Substituting (55) into (54) and separating out terms proportional to $e^{-i\lambda\tau}$ from those proportional to $e^{i\lambda^*\tau}$ (in light of orthogonality), we obtain four equations:

$$\begin{aligned}
0 &= (D_k - \lambda)a_k + \left[\frac{1}{2}(1-\gamma)\psi_{3-k}^*\psi_k - \nu\right]a_{3-k} + \frac{1}{2}(1+\gamma)\psi_k^2 b_k + \frac{1}{2}(1-\gamma)\psi_{3-k}\psi_k b_{3-k}, \\
0 &= (-D_k^* - \lambda)b_k + \left[\nu - \frac{1}{2}(1-\gamma)\psi_{3-k}\psi_k^*\right]b_{3-k} - \frac{1}{2}(1+\gamma)\psi_k^{*2}a_k - \frac{1}{2}(1-\gamma)\psi_{3-k}^*\psi_k^* a_{3-k},
\end{aligned} \quad (56)$$

where

$$D_k = -\frac{1}{2}\partial_{zz} + (1+\gamma)|\psi_k|^2 - 1 + iv_s\partial_z + \frac{1}{2}(1-\gamma)|\psi_{3-k}|^2. \quad (57)$$

When these equations are written in matrix form (in the basis $a_1, b_1, a_2, b_2$), it becomes clear that solving for the $\lambda$'s reduces to diagonalizing the following matrix:

$$M = \begin{pmatrix}
D_1 & \frac{1}{2}(1+\gamma)\psi_1^2 & \frac{1}{2}(1-\gamma)\psi_2^*\psi_1 - \nu & \frac{1}{2}(1-\gamma)\psi_2\psi_1 \\
-\frac{1}{2}(1+\gamma)\psi_1^{*2} & -D_1^* & -\frac{1}{2}(1-\gamma)\psi_2^*\psi_1^* & \nu - \frac{1}{2}(1-\gamma)\psi_2\psi_1^* \\
\frac{1}{2}(1-\gamma)\psi_2\psi_1^* - \nu & \frac{1}{2}(1-\gamma)\psi_2\psi_1 & D_2 & \frac{1}{2}(1+\gamma)\psi_2^2 \\
-\frac{1}{2}(1-\gamma)\psi_2^*\psi_1^* & \nu - \frac{1}{2}(1-\gamma)\psi_1\psi_2^* & -\frac{1}{2}(1+\gamma)\psi_2^{*2} & -D_2^*
\end{pmatrix}. \quad (58)$$

$M$ is a matrix of operators, each of which must also be represented by a matrix. Let us consider these constituent operators first. These operate on the spatial dimension, discretized in steps of $h$. If the interval $[-L, L]$ is discretized in to $N$ grid points, then $\nu$ appearing in $M$ is in fact $\nu$ multiplied by the $N \times N$ identity matrix. The wavefunctions, in turn, are represented by $N \times N$ matrices with $\psi$ on the main diagonal. Products of $\psi$'s are achieved by multiplying the appropriate $\psi$ matrices together.

To construct $\partial_z$ and $\partial_{zz}$ we use a five-point stencil. In particular, if $f(x)$ is some function and $x$ is discretized in steps of $h$, then the first and second derivatives are approximated as

$$f'(x) = \frac{-f(x+2h) + 8f(x+h) - 8f(x-h) + f(x-2h)}{12h},$$

$$f''(x) = \frac{-f(x+2h) + 16f(x+h) - 30f(x) + 16f(x-h) - f(x-2h)}{12h^2}. \tag{59}$$

Thus, the matrices representing the first and second derivative operators only have 5 non-zero diagonals (symmetrically about the main diagonal) which contain the numbers (going from upper-most to lowest diagonal) $\{-1, 8, 0, -8, 1\}/(12h)$ for the first- and $\{-1, 16, -30, 16, -1\}/(12h^2)$ for the second-derivatives. In order to avoid boundary effects, on the second and pre-last rows we use a three point stencil:

$$
\begin{aligned}
f'(x) &= \frac{f(x+h) - f(x-h)}{2h}, \\
f''(x) &= \frac{f(x+h) - 2f(x) + f(x-h)}{h^2}.
\end{aligned} \tag{60}
$$

On the first and last rows, we also use the three point stencil but with additional assumptions. For the first derivative, we are forced to take a one-sided derivative, and for the second derivative, assume that $f(x+h) = f(x-h)$. This is because only one of $x \pm h$ is part of the discrete grid when $x$ is the first or the last point.

To find out whether a solution is stable or not, we need to know if there are any complex eigenvalues. The accuracy of the calculation is limited by $h$, and in our case, $h = L/100$ where $2L$ is the size of the system. $h$ is usually 0.01, but for the largest systems can get up to 0.05 or 0.06. Note that the coupled Gross-Pitaevskii equations in this discrete representation are satisfied to order $h^2$: the norm of the residuals is of order $10^{-4}$. In light of this, the cut-off for deciding whether the complex part of an eigenvalue is spurious or real is set to 0.01. Then, for each complex eigenvalue, the mod-squared eigenvector is inspected. If it is peaked in $[-L/2, L/2]$, it is assumed to be an actual unstable mode. If it peaks outside this range, the complex eigenvalue is assumed to be spurious.

In the $\gamma < 1$ parameter regime, some extra care has to be taken when computing stability. For dark solitons, spurious unstable modes sometimes satisfy our conditions for true instability defined in the paragraph above. To distinguish them from real unstable modes, we required the eigenvector mod-squared at $\pm L$ to have decayed to one hundredth of the maximum value or less. The spurious modes have undamped oscillations beyond the region where the dark soliton is localized and are therefore ruled out by this extra condition. The next issue occurs for both dark solitons and staggered solitons: when the eigenvalue of a true unstable mode goes to zero as a function of some parameter, at some point it inevitably crosses our threshold of 0.01 (set in the paragraph above). This was suspected to occur in the high velocity limits. Therefore we checked that the pure imaginary eigenvalue belonging to the only potentially unstable eigenvector decayed smoothly as a function of velocity to zero. This confirmed that the mode in question was indeed unstable, even though its imaginary eigenvalue was less than 0.01.

## A.1 Analytical stability for dark solitons

We are able to analytically determine the boundary between the stable and unstable regions in parameter space for the known dark soliton solutions. This calculation is not completely general, as in order for it to work, we are forced to assume that the dark solitons are stationary, thus fixing one of the parameters; $v$ and $\gamma$ remain arbitrary, though.

We recall that for dark solitons, $\psi = \psi_1 = \psi_2$ given by (16). Numerically, one finds that the variations of the wavefunctions from (53) always satisfy $\delta\psi = \delta\psi_1 = -\delta\psi_2$, or equivalently, $a = a_1 = -a_2$ and $b = b_1 = -b_2$ (see (55)). Using this knowledge, we can reduce the $4 \times 4$ matrix (58) to a $2 \times 2$ matrix operating on $\vec{\ell} = [a,\ b]^T$:

$$M = \begin{pmatrix} \bar{D} & \gamma\psi^2 \\ -\gamma\psi^{*2} & -\bar{D}^* \end{pmatrix}, \tag{61}$$

where

$$\bar{D} = -\frac{1}{2}\partial_{zz} + i v_s \partial_z + (1+\gamma)|\psi|^2 - 1 + \nu. \tag{62}$$

Numerically we observe that the unstable eigenvector for dark solitons always has zero real part, and therefore, when dark solitons change stability (*i.e.* when the imaginary part of the eigenvalue goes through zero), the entire eigenvalue is zero. We are thus interested in solving $M\vec{\ell} = \vec{0}$. Defining the change of basis matrix

$$U = \begin{pmatrix} 1 & 1 \\ 1 & -1 \end{pmatrix}, \tag{63}$$

we transform our matrix equation into the new basis: $UMU^{-1}U\vec{\ell} = U\vec{0}$, where

$$UMU^{-1} = \frac{1}{2}\begin{pmatrix} \bar{D} - \bar{D}^* + \gamma\psi^2 - \gamma\psi^{*2} & \bar{D} + \bar{D}^* - \gamma\psi^2 - \gamma\psi^{*2} \\ \bar{D} + \bar{D}^* + \gamma\psi^2 + \gamma\psi^{*2} & \bar{D} - \bar{D}^* - \gamma\psi^2 + \gamma\psi^{*2} \end{pmatrix}, \tag{64}$$

and we will denote $U\vec{\ell} = [\tilde{a},\ \tilde{b}]^T$. The choice $v_s = 0$ guarantees that $\psi^2 = \psi^{*2}$ and $\bar{D} = \bar{D}^*$, and hence the diagonal elements of (64) vanish. The resulting equations read

$$\begin{aligned}
0 &= \left[-\frac{1}{2}\partial_{zz} + (1+2\gamma)\psi^2 - 1 + \nu\right]\tilde{a}, \\
0 &= \left[-\frac{1}{2}\partial_{zz} + \psi^2 - 1 + \nu\right]\tilde{b}, \\
\psi &= \sqrt{1+\nu}\tanh\left(\sqrt{1+\nu}z\right).
\end{aligned} \tag{65}$$

These equations have the same form as the (time-independent) Schrödinger equation, *i.e.* the eigen-problem for the Hamiltonian. In addition to the usual kinetic term we have a sech$^2$ potential – known as the Rosen-Morse potential after the authors who first solved this problem analytically [59], and a constant term which can be interpreted as the eigenvalue. The energy spectrum consists of a few discrete bound states (localized and square-integrable), followed by a continuum of higher-energy, unbound states (delocalized). When the parameters are just right for the bound energy eigenvalues of the Hamiltonians to match the eigenvalue terms in the equations, the two equations (65) are satisfied with localized solutions. In other words, for such parameter values a zero eigenvalue of (61) exists and dark solitons switch stability.

Reference [59] derives the following results: the equation

$$\left[\partial_{zz} + \kappa\ \text{sech}^2(z)\right]\psi = \epsilon\psi \tag{66}$$

has discrete, bound eigenvalues

$$\epsilon_n = \left(\sqrt{\kappa + \frac{1}{4}} - n - \frac{1}{2}\right)^2, \tag{67}$$

where $n = 0$ or $n \in \mathbb{N}$, $n \leq \sqrt{\kappa + \frac{1}{4}} - \frac{1}{2}$.

For direct comparison of (65) with this result, we must rewrite the potential terms through $\text{sech}^2$ and change to the scaled position coordinate $\tilde{z} = \sqrt{1+v}z$. This procedure yields

$$\frac{4[\nu + \gamma(1+\nu)]}{1+\nu}\tilde{a} = \left[\partial_{\tilde{z}\tilde{z}} + 2(1+2\gamma)\,\text{sech}^2(\tilde{z})\right]\tilde{a},$$
$$\frac{4\nu}{1+\nu}\tilde{b} = \left[\partial_{\tilde{z}\tilde{z}} + 2\,\text{sech}^2(\tilde{z})\right]\tilde{b}. \qquad (68)$$

Examining the equation for $\tilde{b}$ and comparing to the Rosen-Morse results, $n$ can only be 0 or 1. Moreover, we easily compute $\epsilon_0 = 1$ and $\epsilon_1 = 0$. Next we set each $\epsilon_n$ equal to the eigenvalue $\frac{4\nu}{1+\nu}$ and see what conditions this imposes on our parameters. Doing this for $\epsilon_1$ leads to $\nu = 0$ and for $\epsilon_0$ leads to $\nu = 1/3$. These are well-known points at which dark solitons do change stability: at $\nu = 0$ Josephson vortices appear and dark solitons change from stable to unstable while the reverse process occurs at $\nu = 1/3$.

Now let us compare the equation for $\tilde{a}$ to the Rosen-Morse results: $n$ can be 0, 1 or 2, the latter only if $\gamma \geq 5/8$. Setting $\epsilon_n$ equal to the eigenvalue of the $\tilde{a}$ equation gives

$$\epsilon_n = \left(\sqrt{2(1+2\gamma) + \frac{1}{4}} - n - \frac{1}{2}\right)^2 = \frac{4[\nu + \gamma(1+\nu)]}{1+\nu}. \qquad (69)$$

We can use this condition to check our numerical results. Setting $\nu = 0.005$, and taking $n = 0, 1, 2$ in turn, we plot the left- and right-hand sides of (69) as a function of $\gamma$, looking for the intersection point. For $n = 0$ (69) is satisfied at $\gamma \approx 0.975$, for $n = 1$ the lines do not cross and for $n = 2$ they cross at $\gamma \approx 0.1565 < 5/8$, so $n = 2$ is not actually possible at this point in parameter space. Thus, this calculation predicts that stationary dark solitons at $\nu = 0.005$ will change stability once, at $\gamma \approx 0.975$. This point is added to Fig. 10 (red square) and fits perfectly on the numerical curve (blue circles).

## B Derivation of the sine-Gordon equation

In this appendix we show how one can obtain the sine-Gordon equation from the Gross-Pitaevskii model of section 2. The Lagrangian density of the coupled BECs system is given by

$$\mathscr{L} = \mathscr{L}_B - w, \qquad (70)$$

where the energy density (also see (9)) is

$$w = \sum_k \left\{ \frac{\hbar^2}{2m}|\partial_x \Psi_k|^2 - \mu |\Psi_k|^2 - J\Psi_k^* \Psi_{3-k} + \frac{1}{2}g|\Psi_k|^4 \right\} + g_c |\Psi_1|^2 |\Psi_2|^2, \qquad (71)$$

and

$$\mathscr{L}_B = \frac{i\hbar}{2}\sum_k \left(\Psi_k^* \partial_t \Psi_k - \Psi_k \partial_t \Psi_k^*\right). \qquad (72)$$

The Gross-Pitaevskii equations (2) can be recovered from the Euler-Lagrange equations for the fields $\Psi_k$ and $\Psi_k^*$. To proceed, we take the following ansatz for the wavefunctions:

$$\Psi_1(x,t) = u(x,t)\cos[\Theta(x,t)]e^{\frac{i}{2}[\phi_s(x,t)+\phi_a(x,t)]},$$
$$\Psi_2(x,t) = u(x,t)\sin[\Theta(x,t)]e^{\frac{i}{2}[\phi_s(x,t)-\phi_a(x,t)]}. \qquad (73)$$

In terms of the new fields, (70) becomes

$$
\begin{aligned}
\mathscr{L} = & -\frac{\hbar}{2}u^2\left[\partial_t\phi_s + \cos(2\Theta)\partial_t\phi_a\right] - \frac{\hbar^2}{2m}\Big\{(\partial_x u)^2 + u^2(\partial_x\Theta)^2 \\
& + \frac{u^2}{4}\left[(\partial_x\phi_s)^2 + (\partial_x\phi_a)^2\right] + \frac{u^2}{2}\cos(2\Theta)\partial_x\phi_s\partial_x\phi_a\Big\} \\
& + \mu u^2 + Ju^2\sin(2\Theta)\cos(\phi_a) - \frac{g}{2}u^4\left[\cos^4(\Theta) + \sin^4(\Theta)\right] \\
& \hspace{6cm} - g_c u^4\cos^2(\Theta)\sin^2(\Theta). \quad (74)
\end{aligned}
$$

We now assume that the densities of the two wavefuncitons are almost the same, *i.e.*, we take

$$
\Theta(x,t) = \frac{\pi}{4} + y(x,t), \quad (75)
$$

where $y$ is a field of small magnitude. We expand $\mathscr{L}$ to second order in $y$:

$$
\begin{aligned}
\mathscr{L} = & -\frac{\hbar}{2}u^2\left[\partial_t\phi_s - 2y\,\partial_t\phi_a\right] - \frac{\hbar^2}{2m}\Big\{(\partial_x u)^2 + u^2(\partial_x y)^2 \\
& + \frac{u^2}{4}\left[(\partial_x\phi_s)^2 + (\partial_x\phi_a)^2\right] - yu^2\partial_x\phi_s\partial_x\phi_a\Big\} \\
& + \mu u^2 + Ju^2(1-2y^2)\cos(\phi_a) - \frac{g}{4}u^4(1+4y^2) \\
& \hspace{6cm} - \frac{g_c}{4}u^4(1-4y^2). \quad (76)
\end{aligned}
$$

Expanding out all the brackets in (76), we keep only the $2^{\text{nd}}, 6^{\text{th}}, 9^{\text{th}}, 12^{\text{th}}$, and $14^{\text{th}}$ terms. This selection is based upon whether or not the term is needed in the reduced Lagrange density in order for it to yield the sine-Gordon equation. The reduced Lagrangian reads

$$
\mathscr{L} = \hbar y u^2\partial_t\phi_a - \frac{\hbar^2}{2m}\frac{u^2}{4}(\partial_x\phi_a)^2 + Ju^2\cos(\phi_a) - u^4(g-g_c)y^2. \quad (77)
$$

We write down the Euler-Lagrange equations for $y$ and $\phi_a$, make the approximation that $u$ is a constant, eliminate $y$ between the two equations and get

$$
\partial_{tt}\phi_a - \frac{\gamma}{m}(\mu+J)\partial_{xx}\phi_a = -\frac{4\gamma(\mu+J)}{\hbar^2}\sin(\phi_a), \quad (78)
$$

where $u$ was set to the background value,

$$
u = \sqrt{2\frac{\mu+J}{g+g_c}}. \quad (79)
$$

Equation (38) is identical to (78).

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
