# Peer review of "Quasiparticles of widely tuneable inertial mass: The dispersion relation of atomic Josephson vortices and related solitary waves"

_SciPost Physics, doi:SciPost Phys. 4, 018 (2018)_

## Round 1 · Referee Report · Anonymous (Referee 1) · 2017-9-26

Strengths

1 - Good introduction and review of the previously known results;
2 - The motivation of the investigation is clearly explained.
3 - The calculation and results seem to be on solid ground.

Weaknesses

1 - The presentation is not well organized.
2 - The figures need to be improved for a better view of the results.

Report

This manuscript studies solitary waves in Rabi-coupled Bose-Einstein condensates. The authors first reviewed the known soliton solutions, including dark solitons, stationary Josephson vortex and Manakov solitons in this Rabi-coupled system. Then they numerically solved the problem and found moving Josephson vortex solutions. A full dispersion relation (i.e, the relation between soliton’s energy and its velocity) is mapped out. A striking property is that the effective mass of the moving soliton could change sign at a critical point. The property and the stability of the soliton are also explored for a wide range of Rabi-coupling strength and interaction constants.

I have two major comments on the current manuscript:

(1) The discussion of the figures should be improved. For some of the figures, there is only a very brief explanation in the main text which makes the manuscript difficult to read.

I have to jump between different sections to understand the same figure of this manuscript. Since there are so many figures in the manuscript, this should be avoided as much as possible.

It is also highly recommended to replace Fig3 and Fig7 by 1D plots. The transverse direction is obviously not important at all. It is difficult to see the profile of solitons in these 2D plots.

(2) Around Eq.(21), the authors mentioned Ref[28] when they discussed the stationary Josephson vortex. If I understand it correctly, when g_c is very close to g, that is when \gamma<<1, the domain wall found in Ref[28] is approximately the same as the stationary Josephson vortex (The approximation comes from the assumption that n1+n2 is a constant in Ref[28]).

Based on this consideration, the recently studied magnetic soliton (see PRA 95, 033614 (2017)) seems to coincide with the moving JV solution investigated by the authors in the limit of \gamma<<1. For example, Fig7(d) looks very similar to the so-called "magnetic soliton" introduced in the above PRA paper. Furthermore, the effective mass of magnetic soliton is also widely tunable and can change sign.

Is it correct that the results of this PRA paper are a special case of the general results presented in this manuscript? The authors need to clarify the similarities and differences between their work and this reference.

A minor comment:

- On page 7, "In the scenario where the two components are spatially separated it has the significance of an orbital angular momentum". The referee is not very clear of this statement. Is this only because it is possible to formally define an angular momentum L=r\times p where r is the vector pointing from one well to the other? Is there any physical consequence of this quantity?

Requested changes

1 - Improve figures and the corresponding discussions.
2 - Clarify the difference and relation of their results to a know solution (see the Report)

---

## Round 1 · Referee Report · Anonymous (Referee 2) · 2017-10-27

Strengths

1) a comprehensive review of solitonic solutions in two-component coherently coupled Bose Einstein Condensates;
2) calculations (both analytical and numerical) covering a wide range of parameters;
3) description of numerical procedure;
4) detailed, high-quality figures;
5) well written;

Weaknesses

in the report

Report

The article "Quasiparticles of widely tuneable inertial mass: The dispersion relation of atomic Josephson vortices and related solitary waves" constitutes a very comprehensive review of possible solitonic solutions in two-component coherently coupled Bose Einstein Condensates (BECs). The main aim of the article is to investigate a variety of solitonic solutions moving with a constant velocity v_s: dark solitons, the so-called Josephson Vortices (hereafter JVs, also known as domain walls as in Ref. [28]) and a new class of defects called staggered solitons. This is achieved by solving the coupled Gross-Pitaevskii equations (GPEs), Eq. (6) and calculating a variety of quantities: energy of the excitations, canonical momentum, inertial mass and the particle number depletion. The Authors also analyze the linear stability of the solutions they find (details are discussed in the appendix), and show how they bifurcate when the coherent coupling strength is varied.

The paper starts with a review of the literature on the subject. Then in Sec. 2 the Authors introduce the equations defining the system and dimensionless variables they will use throughout the paper. It is followed by a review of analytically known solutions: dark solitons and stationary JVs, and a wide class Manakov solitons parametrized by a phase factor (theta). One of the strengths of the article is that the Authors give a description of applied numerical procedures; they solve Eqs (6) as an open boundary value problem. The text is complemented by visualizations of density and phase of particular, representative solutions.

The central results are summarized in Fig. 2, which shows calculated dispersion relations and in Figs. 4, 5, and 6, where "phase diagrams" (energy and momentum vs velocity of a soliton) are shown. As the value of the coherent coupling is varied, the dispersion relation of the JV changes continuously from a single maximum curve to a function with a minimum at the centre and two maxima on its sides. Therefore, the inertial mass of the defect corresponding to this central point changes sign from positive to negative while the coupling strength is being lowered. The Authors identify the critical value of the linear coupling, at which the inertial mass diverges. The numerical calculations are further supported by a variational calculation for JVs, where the used ansatz is similar to the Manakov solution, Eq. (26), with some parameters to be adjusted. The variational calculation is performed assuming small velocities (v_s), and the parameters are linearized in v_s. Thus calculated inertial mass is plotted together with other results in Figs. 11 and 13, and it gives a surprisingly good estimate.

Finally, Authors derive the sine-Gordon equation (SGE, derivation in the appendix) as an approximation to GPEs , Eqs. (6), in small coupling limit. They do it by dropping certain terms such that the result agrees with their initial assumption. This is, in my opinion, the controversial point of the manuscript (see questions below). The Authors conclude, that the approximation by SGE is valid only for small values of the coherent coupling. In that region, it well approximates the minimum of the dispersion relation for JVs by a parabola, and therefore it can be interpreted as a relativistic description of the excitations (since E2sp2+m2). The authors conclude that "by working in the small ν regime, Josephson vortices may open the possibility for experimental study of 'relativistic particles' (to a good approximation) using collective excitations of ultracold atoms".

I think the article presents a valuable contribution as well as a comprehensive review on the subject, and it definitely merits a publication. However, I would suggest to consider my questions (listed below) before the final publication.

1) The manuscript deals with two-component linearly coupled BECs, where a variety of phases might be present (without solitonic excitations). I think the Authors should comment how the parameters regimes they consider correspond to the miscible, immiscible, polarized or unpolarized phases.

2) In the variational calculation of Sec. 10, the inertial mass is expanded in velocity v_s, but the missing particle number is taken from the zero velocity solution (and the ratio of the two is shown in Fig. 13). Why is it only the mass that has to be expanded?

3) In their derivation of the SGE, the Authors write: "In addition, many terms are dropped from the Lagrangian density based on the fact that the remaining terms yield the sine-Gordon equation (this selection is partly justified a posteriori by the success of the analysis we perform in section 12)". Can this omission of certain terms be explained in terms of the Renormalization Group flow (and the dropped terms as corresponding to the irrelevant operators)?

4) In the conclusion, the Authors write: "If one worked in the bistable region, however, since dark solitons are stable they would not decay. This could potentially enable one to observe dynamics and interaction of Josephson vortices with dark solitons experimentally." This does not exclude, however, a possible thermodynamical instability of solutions from higher energy branches. Should not this be taken into account as well when considering the experiments performed in small but finite temperature?

In summary, I could recommend publication of the manuscript in SciPost, but I would be curious to read the Authors' response to my comments first.

Requested changes

1) The manuscript deals with two-component linearly coupled BECs, where a variety of phases might be present (without solitonic excitations). I think the Authors should comment how the parameters regimes they consider correspond to the miscible, immiscible, polarized or unpolarized phases.

2) In the variational calculation of Sec. 10, the inertial mass is expanded in velocity v_s, but the missing particle number is taken from the zero velocity solution (and the ratio of the two is shown in Fig. 13). Why is it only the mass that has to be expanded?

3) In their derivation of the SGE, the Authors write: "In addition, many terms are dropped from the Lagrangian density based on the fact that the remaining terms yield the sine-Gordon equation (this selection is partly justified a posteriori by the success of the analysis we perform in section 12)". Can this omission of certain terms be explained in terms of the Renormalization Group flow (and the dropped terms as corresponding to the irrelevant operators)?

4) In the conclusion, the Authors write: "If one worked in the bistable region, however, since dark solitons are stable they would not decay. This could potentially enable one to observe dynamics and interaction of Josephson vortices with dark solitons experimentally." This does not exclude, however, a possible thermodynamical instability of solutions from higher energy branches. Should not this be taken into account as well when considering the experiments performed in small but finite temperature?

---

## Round 2 · Author Response

First of all, we would like to thank both referees for the time and effort they have invested in reviewing our manuscript, as well as the constructive feedback and interesting questions. Please find below a response to all points raised by the referees and a summary of the changes made.

Referee 1:

(1) The referee comments that discussion of the figures should to be improved and should occur near where the figures are placed in the manuscript. We have thus moved old Fig. 2 (now Fig. 5) and Figs. 4-6 (now Figs. 6-8) to more relevant places in the manuscript. We have also made small adjustments to the discussion of the figures in the text and, in particular, have added cross-references in the text where figures are (first) mentioned to the section where the main discussion of the figures is located (this is relevant also to the old Figs. 11 and 13, which are now Figs. 12 and 14).

Further, the referee suggests replacing old Figs. 3 & 7 by one-dimensional plots rather than iso-surfaces. We do quite like the 3D plots, which encode the full information (phase in the colour and density in the transverse extent of the iso-surface), for their intuitive appeal but have decided to show them alongside more conventional line plots in the new figures 2, 3, and 4 (previously 3 and 7).

(2) The referee points out the article Magnetic solitons in Rabi-coupled Bose-Einstein condensates by Qu et al. Indeed this paper is highly relevant and we thank the referee for pointing it out. We only became aware of it after submitting our manuscript for publication. We have now incorporated this reference into the literature review in the introduction.

Finally, the referee is correct in their interpretation of the meaning of the angular momentum ΔP when the two condensates are in a double-well potential. This is a useful quantity because only the Josephson vortices have a non-zero ΔP, while dark and staggered solitons do not. In fact, the property of ΔP≠0 can be used to distinguish vortex and soliton excitations. A note to this end has been added at the end of section 7.

Referee 2:

The referee poses four interesting questions, to which we now reply:

1) The miscible/immiscible threshold in our system occurs at γ=0 (previously mentioned in passing under equation (51)), while the polarised/unpolarised threshold at υ=0. For υ>0 the ground state has equal densities in the two components and, in spinor notation, is proportional to the vector (1,1). Since this is an eigenstate of the σx operator, the state is polarised in the x-direction. This is reflected in the fact that the phases of the two condensates are linked: they must be pairwise equal at each edge of the box. This is no longer the case if υ=0, when the phases of the two strands become completely independent. Moreover, all states on the equator of the Bloch sphere are then degenerate, making for a degenerate mean-field ground state. We constrained our investigation to the regime where the ground state is miscible (γ>0) and polarised (υ>0). A comment to this effect has been added at the start of section 4 where the background solution is introduced.

2) Regarding the magenta dash-dotted curve in Fig. 14 (old Fig. 13), we have indeed used the variational solution for the Josephson vortex around the centre of the dispersion relation to compute mI. Finite velocity solutions are necessary because mI is defined as a derivative with respect to velocity of the dispersion, as per equation (13), but once the derivative is taken, it is evaluated at zero velocity. The missing particle number, on the other hand, can be computed directly for the stationary vortex, with no need to call upon the approximate moving solutions.

We have added a short note specifying that the plots pertain to zero velocity in the captions of Figs. 12, 13, 14 (old Figs. 11, 12, 13) to eliminate confusion.

3) We do not believe that there is a simple explanation in terms of renormalisation group flow due to the presence of another (gapless) excitation sector, which is absent in the sine-Gordon equation. These two sectors asymptotically decouple in the limit υ→0, but since we do not have strong analytical arguments to justify our term selection on this basis, we prefer to keep the current presentation based on a posteriori justification. A comment clarifying the situation has been added in section 11.

4) The referee raises the excellent question of the stability of our excitations in the presence of a thermal cloud. Of course all ultra-cold atom experiments are done at a finite temperature, and it is well-known that over a long period of time, collisions between the dark soliton and the thermal atoms lead to the decay of the soliton. However, this process is comparatively slow, allowing one to observe the soliton for prolonged periods of time. This is due to the fact that the dark soliton possesses a macroscopic feature in the form of a phase step, spanning the entire cloud, so momentum exchange between the condensate and thermal atoms is suppressed. Precisely such collisions, however, are needed in order to allow the phase to progressively unwind, breaking translational invariance, speeding up the soliton until it eventually disappears at the speed of sound. Note that this occurs due to the concave-down dispersion relation of dark solitons.

On the other hand, in the regime where the Josephson vortex branch possesses a local minimum in the dispersion relation, vortex excitations around this minimum should be stabilized by a thermal cloud. Overall, we think it should still be possible to see both dark solitons and Josephson vortices in the regime of bistability, however the lifetime of any excitation with a negative inertial mass will be shortened by the thermal cloud. We have added a brief comment in this spirit at the end of section 13.

---

## Editorial Decision

published